# Grainyhead 1 acts as a drug-inducible conserved transcriptional regulator linked to insulin signaling and lifespan

Giovanna Grigolon [1,5], Elisa Araldi[1,2], Reto Erni [3], Jia Yee Wu [1], Carolin Thomas[1], Marco La Fortezza [4], Beate Laube[1], Doris Pöhlmann[1], Markus Stoffel [2], Kim Zarse [1], Erick M. Carreira [3], Michael Ristow [1,6 ✉] & Fabian Fischer [1,5,6]

Aging is impacted by interventions across species, often converging on metabolic pathways. Transcription factors regulate longevity yet approaches for their pharmacological modulation to exert geroprotection remain sparse. We show that increased expression of the transcription factor Grainyhead 1 (GRH-1) promotes lifespan and pathogen resistance in *Caenorhabditis elegans*. A compound screen identifies FDA-approved drugs able to activate human GRHL1 and promote nematodal GRH-1-dependent longevity. GRHL1 activity is regulated by post-translational lysine methylation and the phosphoinositide (PI) 3-kinase C2A. Consistently, nematodal longevity following impairment of the PI 3-kinase or insulin/IGF-1 receptor requires *grh-1*. In BXD mice, *Grhl1* expression is positively correlated with lifespan and insulin sensitivity. In humans, *GRHL1* expression positively correlates with insulin receptor signaling and also with lifespan. Fasting blood glucose levels, including in individuals with type 2 diabetes, are negatively correlated with *GRHL1* expression. Thereby, GRH-1/GRHL1 is identified as a pharmacologically malleable transcription factor impacting insulin signaling and lifespan.

[1] Energy Metabolism Laboratory, Institute of Translational Medicine, Department of Health Sciences and Technology, Swiss Federal Institute of Technology (ETH) Zurich, Schwerzenbach CH-8603, Switzerland. [2] Metabolism and Metabolic Disease Laboratory, Institute for Molecular Health Sciences, Department of Biology, Swiss Federal Institute of Technology (ETH) Zurich, Zurich CH-8093, Switzerland. [3] Laboratory of Organic Chemistry, Department of Chemistry and Applied Biosciences, Swiss Federal Institute of Technology (ETH) Zurich, Zurich CH-8093, Switzerland. [4] Evolutionary Biology Laboratory, Department of Environmental Systems Science, Swiss Federal Institute of Technology (ETH) Zurich, Zurich CH-8092, Switzerland. [5] These authors contributed equally: Giovanna Grigolon, Fabian Fischer. [6] These authors jointly supervised this work: Michael Ristow, Fabian Fischer. ✉ email: michael-ristow@ethz.ch

Research of the last decades provides compelling evidence to support the notion that aging, including that of humans, can to some extent be modulated. Life- and healthspan of different organisms, such as *Caenorhabditis elegans*, *Drosophila melanogaster*, and mice, are extendable by dietary[1], genetic[2], and pharmacological interventions[3], often converging on metabolic pathways[4]. To develop pharmacological treatments to counteract aging-associated diseases[5,6] and the general functional decline during in particular human aging, one promising recent approach is the identification of evolutionarily conserved and pharmacologically modulable targets controlling the aging process across species[7]. This strategy allows the use of experimentally accessible cell culture models and comparatively simple animal model organisms for the rational selection of novel lead compounds and repurposed drugs to potentially serve as geroprotectors, before transitioning toward more complex, costly, and time-consuming preclinical studies and first-in-human trials.

Aberrations in transcriptional networks have been identified as early drivers of aging and one of its potential root causes[8]. Reflecting this, transcription factors (TFs), many of which have been shown repeatedly and with remarkable evolutionary consistency to control development[9], metabolism[10], and stress responses[11], are also increasingly linked to regulation of longevity[12]. Some TFs are demonstrated to impact lifespan of model organisms in a conserved manner and, at least by correlation, already implied to contribute to human longevity, as perhaps currently best represented by Forkhead box subfamily O (FOXO) TFs[13], as well as the nuclear factor erythroid 2-related factor 2 (NFE2L2 aka NRF2)[14].

In a previous study from our laboratory, impairment of Grainyhead 1 (GRH-1) in *C. elegans*, the sole nematodal orthologue of mammalian Grainyhead-like (GRHL) TFs[15,16], was identified to shorten lifespan[17]. We now show that limited overexpression of *grh-1* promotes nematodal lifespan without impairing motility or fertility, while also improving pathogen resistance of *C. elegans*. Small-molecule compounds, including the FDA-approved phytochemical papaverine, are identified as human GRHL1 activators, capable of extending nematodal lifespan through GRH-1. Human GRHL1 activity is here shown as controlled by lysine dimethylation and regulated by a methylase (KMT2D; SET-16 in *C. elegans*), demethylase (KDM1A; LSD-1 and SPR-5 in *C. elegans*), and PI3 kinase (PIK3C2A; PIKI-1 in *C. elegans*) in response to papaverine treatment. Consistent with PI3Ks being involved in insulin/IGF-1 signaling (IIS)[18], we observe that nematodal *grh-1* is necessary for longevity upon IIS impairment.

Supporting an evolutionarily conserved link of Grainyhead transcription factors to aging and IIS, we find *Grhl1* expression in the BXD mouse genetic reference population[19] to positively correlate with increased lifespan and improved insulin sensitivity. Furthermore, we find that *GRHL1* expression in humans correlates with insulin receptor signaling and, similar to what is observed in BXD mice, that its increased expression predicts prolonged adult lifespan. In addition, human *GRHL1* expression negatively correlates with fasting blood glucose levels, notably also including in individuals with type 2 diabetes.

Together, these results identify GRH-1/GRHL1 as a drug-inducible aging-related TF linked to regulation of insulin signaling and longevity and that substances activating GRHL1, such as papaverine, could potentially serve as lead compounds for the development of geroprotective drugs.

## Results

### Grainyhead 1 transcription factor regulates aging of nematodes.
GRHL TFs are conserved from fungi to mammals, with mice and humans possessing three functionally distinct

orthologues (Grhl/GRHL1-3) and nematodal GRH-1 most closely related to Grhl1/GRHL1[15,16]. Post-developmental impairment of *C. elegans grh-1* in nematodes by RNA interference (RNAi) significantly shortened lifespan (mean −17.0%, $P < 0.001$; Fig. 1a), as previously described[17]. On average, in three experiments, mean lifespan with *grh-1* RNAi was reduced by 17.3 ± 10.9% compared to feeding with the L4440 control vector. Notably, lifespan of the *grh-1* deletion strain VC2072 (Δ*grh-1*), carrying the *grh-1* frameshift deletion allele gk960[20], was not significantly different compared to wild-type N2 nematodes (WT) (Fig. 1b). See Supplementary Data 1 for detailed statistics of all lifespan assays and repeats thereof throughout this study.

Given these divergent results following *grh-1* impairment vs. deletion and to further test if GRH-1 is indeed involved in the regulation of nematodal lifespan, two independent strains (named line 1 and 2) overexpressing *grh-1::gfp* under control of the heat shock-inducible *hsp-16.2*-promoter (*grh-1* OEx) were generated. Without heat shock induction, *grh-1* overexpression of *grh-1* OEx line 1 was ~2-fold ($P = 0.047$; Fig. 1c), with no detectable nuclear GFP signal (Fig. 1d and Supplementary Fig. 1a), and lifespan of this strain significantly extended (mean + 16.3%, $P < 0.001$; Fig. 1e). This longevity phenotype was confirmed with the independently generated *hsp-16.2p::grh-1::gfp* OEx line 2, which also displayed limited *grh-1* overexpression (~3-fold) and a significantly extended lifespan in absence of heat shock induction (Supplementary Fig. 1b, c). Transient heat shock of *grh-1* OEx line 1 resulted in a stronger ~30-fold *grh-1* overexpression ($P = 0.005$; Fig. 1f), visible nuclear accumulation of GRH-1::GFP (Fig. 1g and Supplementary Fig. 1d), and reduction of lifespan compared to WT nematodes subjected to the same heat shock treatment in parallel (mean −29.6%, $P < 0.001$; Fig. 1h). Similar results were again obtained with the independently generated *grh-1* OEx line 2, which also displayed strongly increased *grh-1* overexpression together with shortened lifespan after heat shock (Supplementary Fig. 1e, f). Considering the overall identical results using these two independent *grh-1* OEx lines in initial lifespan assays, all following experiments were conducted using line 1, hereafter simply referred to as *grh-1* OEx. Treatment of *grh-1* OEx with *grh-1* RNAi abolished longevity, thus indicating that increased *grh-1* is indeed responsible for extending the lifespan of this strain (Supplementary Fig. 1g). In addition, overexpression of *grh-1* under control of its endogenous promoter resulted in an intermediate ~10-fold increase of *grh-1* mRNA and also shortened lifespan (Supplementary Fig. 1h, i), with the effect on lifespan tendentially less pronounced than in the heat shocked *grh-1* OEx lines. Together, this indicates a dose-dependent effect of *grh-1* expression on lifespan, with both its reduction by RNAi and stronger overexpression shortening lifespan, while limited overexpression promotes longevity.

To assess effects of *grh-1* overexpression on overall fitness, the motility and fertility of non-heat shocked *grh-1* OEx nematodes were assayed. Limited *grh-1* overexpression significantly increased maximum speed (Fig. 1i) and did not affect brood size (Fig. 1j). In contrast, inducing *grh-1* overexpression by heat shock treatment abolished positive effects on motility and led to reduced fertility (Fig. 1i, j), underscoring a dose-dependent effect of *grh-1* not only on lifespan but also on health parameters. Given the protective role of limited *grh-1* overexpression, all subsequent experiments were performed with the *grh-1* OEx strain in absence of a heat shock.

### GRH-1 promotes MAPK signaling and pathogen resistance.
RNA-Seq of WT vs. *grh-1* OEx (Supplementary Data 2) revealed MAP kinase activity (GO:0004707) as the most significantly enriched pathway among transcripts increased by *grh-1* overexpression (Supplementary Fig. 2a) and *pmk-1*, encoding a

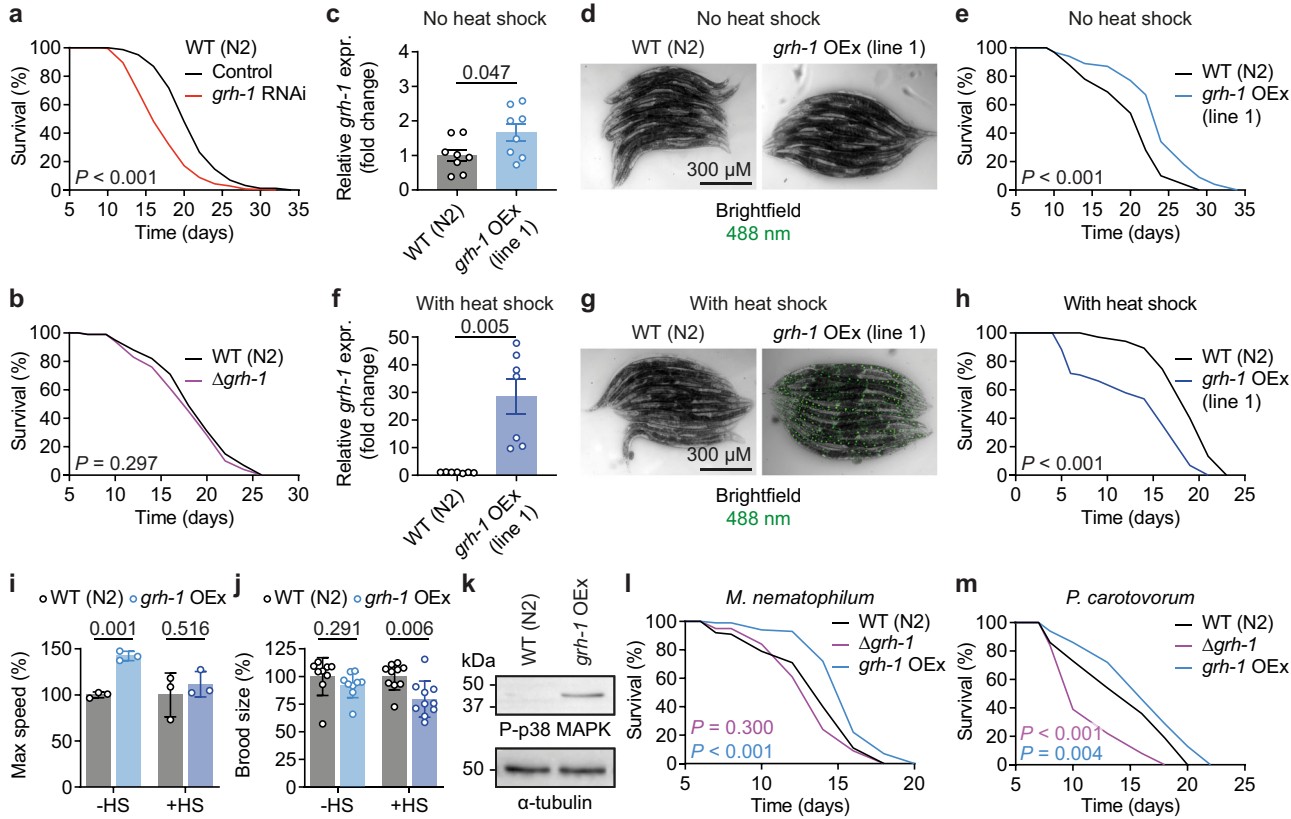

**Fig. 1 GRH-1 is a longevity-promoting transcription factor. a** Lifespan assay of wild-type (WT) N2 nematodes on post-developmental *grh-1* RNAi compared to control vector L4440. **b** Lifespan assay on OP50 bacteria using a *grh-1* deletion strain (null mutant VC2072) vs. WT (N2). **c–e** WT (N2) vs. *grh-1 OEx* line 1 (without heat shock) RT-qPCR of *grh-1* with *n* = 8 for both (**c**), microscopy images to detect GRH-1::GFP (**d**), and lifespan on OP50 bacteria (**e**). **f–h** Same experiments as in (**c–e**) but in each case using heat shocked *grh-1* OEx line 1 vs. respective WT (N2) control (RT-qPCR with *n* = 7 for both). **i, j** Motility (**i**) and fertility (**j**) assays of WT (N2) vs. *grh-1* OEx sans (−HS) or with (+HS) heat shock (motility assay with *n* = 3 for all; fertility assay with *n* = 9 for −HS and *n* = 10 for +HS). **k** Western blot of phospho-p38 MAPK in WT (N2) and *grh-1* OEx with α-tubulin as loading control. **l, m** Lifespan assay of WT (N2) vs. Δ*grh-1* and *grh-1* OEx on pathogenic *M. nematophilum* (**l**) or *P. carotovorum* bacteria (**m**). Data in bar graphs are mean ± SEM, with sample sizes as stated and individual data points representing biological replicates. *P*-values were determined with two-tailed unequal variances *t*-tests of the indicated comparisons of unpaired control vs. treatment groups. *P*-values of *C. elegans* lifespan assays were determined by log-rank test. See Supplementary Data 1 for detailed lifespan assay statistics, Supplementary Fig. 1a, d for microscopy images corresponding to (**d** and **g**), and Supplementary Fig. 2f for full western blot images corresponding to (**k**). Source data are provided as a Source data file.

human p38 MAP kinase orthologue, as the most significantly upregulated gene thereof (log2 ratio = 1.056, $P = 4.3 \times 10^{-10}$). Notably, both *pmk-2* and *pmk-3*, which also belong to the MAP kinase activity pathway, were likewise significantly overexpressed in *grh-1* OEx (Supplementary Data 2). Furthermore, as determined by two independent bioinformatic prediction tools used to scan all genes whose expression was increased in *grh-1* OEx, *pmk-1* was determined to contain GRH-1 binding motifs within its predicted promoter (Supplementary Fig. 2b and Supplementary Data 3), altogether indicating it as a possible direct target gene of GRH-1. Gene set enrichment analysis (GSEA) of human genes positively correlating with human *GRHL1* expression in non-gender specific tissues identified the MAP kinase activity pathway (GO:0004707) as also significantly enriched among them (see Supplementary Fig. 2c for pathways correlating with nematodal *grh-1* and with human *GRHL1* by GSEA). In particular, three out of four human *pmk-1* orthologues (MAPK11, MAPK13, and MAPK14) were found positively correlated with *GRHL1* expression in muscle (Supplementary Fig. 2d), and similar correlations were observed with *GRHL1* in liver (Supplementary Fig. 2e). This indicates an evolutionarily conserved link between Grainyhead TFs and MAPK signaling.

Consistent with the MAPK pathway being regulated by GRH-1, increased levels of phosphorylated p38 MAPK were detected in *grh-1*

OEx compared to WT (Fig. 1k and Supplementary Fig. 2f). In addition, lifespan-extension of *grh-1* OEx was abolished by *pmk-1* RNAi (Supplementary Fig. 2g), indicating this gene as required for mediating longevity of *grh-1* OEx. Consistent with *pmk-1* regulating the *C. elegans* immune response[21,22], *grh-1* OEx conferred resistance against two pathogenic bacterial strains, i.e., *Microbacterium nematophilum* (Fig. 1l) and *Pectobacterium carotovorum* (Fig. 1m). Since a combined lifespan assay of *grh-1* RNAi bacteria together with these pathogenic bacteria was not feasible, the effect of *M. nematophilum* and *P. carotovorum* in absence of *grh-1* was checked instead. When fed with *P. carotovorum*, lifespan of Δ*grh-1* was reduced, both in comparison to WT and *grh-1* OEx (mean −18.1% vs. WT and −27.5% vs. *grh-1* OEx, *P* < 0.001; Fig. 1m).

### A small molecule screen identifies activators of human GRHL1.

Having established Grainyhead 1 as an aging-related TF in nematodes, a screening to identify compounds modulating its activity was performed. For this purpose, a human HEK293 GRHL reporter was generated (GRHL^WT-LUC), in which several repeats of the mammalian GRHL consensus binding sequence[16] control expression of a luciferase reporter gene (Supplementary Fig. 3a, b). GRHL^WT-LUC was validated to be activated by treatment with the histone deacetylase inhibitor (HDACi) Scriptaid

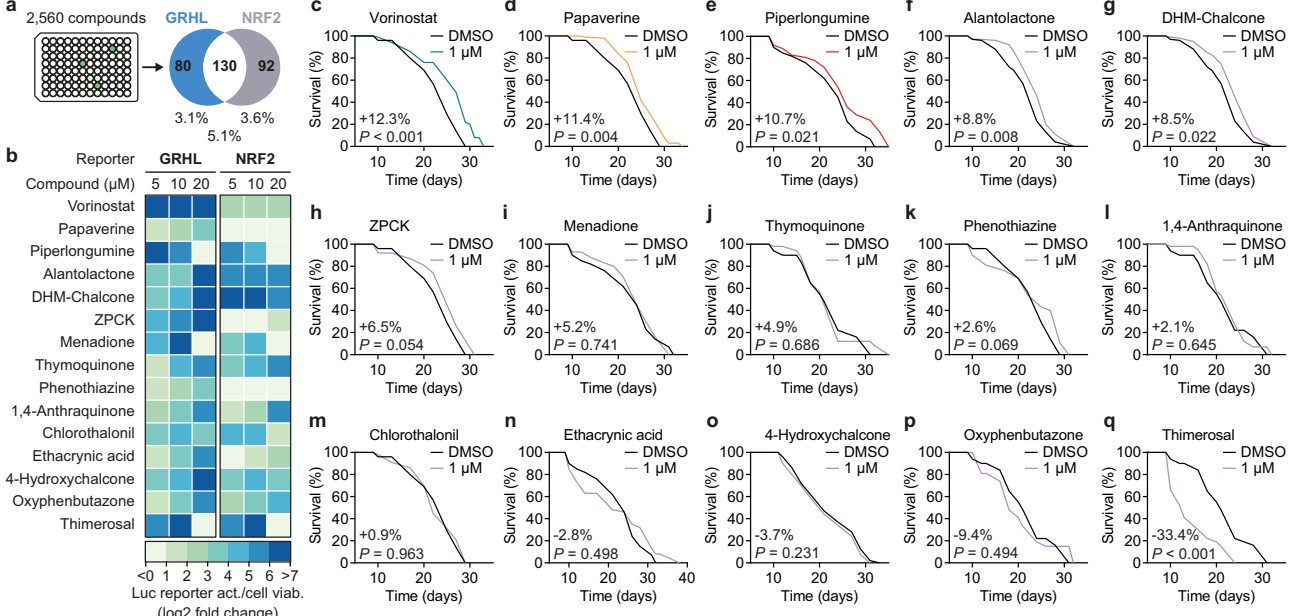

**Fig. 2 Human GRHL1-activating compounds and their impact on nematodal lifespan. a** Schematic Venn diagram of hits activating a human GRHL and/or NRF2 luciferase reporter with fold change ≥2 at 10 μM in the primary screening. Indicated are total number of hits and hits as percentage of full library (2560 compounds). See Supplementary Data 4. **b** Heat map depicting log2 fold change reporter activation in a three-point dose–response assay (5, 10, and 20 μM) of the top 15 GRHL hit compounds on the reporters as in (**a**), normalized to cell viability (*n* = 2–3 per condition). **c–q** Lifespan assays, all using WT (N2) nematodes treated with the respective individual compound at a concentration of 1 μM in plate and vs. DMSO control. Lifespan graphs contain percentage change of mean lifespan compound vs. DMSO control and *P*-values as determined by log-rank test. See Supplementary Data 1 for detailed lifespan assay statistics. Source data are provided as a Source data file.

(63.7 ± 2.9 fold, $P = 8.7 \times 10^{-13}$; Supplementary Fig. 3c), selected as a positive control compound based on *GRHL1* expression being strongly induced by HDACis[23]. Similarly, an independent human HEK293 reporter for the antioxidant response element (ARE) (Supplementary Fig. 3a, b) targeted by the redox-sensitive TF NRF2[24], was generated and confirmed as responsive to the canonical NRF2-activator sulforaphane[25] (7.5 ± 0.4-fold, $P = 2.3 \times 10^{-9}$; Supplementary Fig. 3c). These two independent reporter cell lines were used in parallel for primary screening of a library of 2560 compounds, comprising phytochemicals and approved drugs, at 10 μM final concentration. Thereby, 5.1% of the library (130 compounds) were identified as activators (≥2-fold activation) of both the GRHL and NRF2 reporter, whereas 3.1% (80 compounds) only activated the GRHL and 3.6% (92 compounds) only the NRF2 reporter (Fig. 2a, Supplementary Data 4).

The 15 compounds with the highest activation of GRHL<sup>WT-LUC</sup> that were readily available for resourcing were defined as primary hits and evaluated in more detail (see list of compounds in Fig. 2b). First, a three-point dose–response reporter assay (5, 10, and 20 μM final concentration), in independent repeat experiments and normalized to cell viability, was performed. All 15 primary hits were confirmed as activators of GRHL<sup>WT-LUC</sup> and 13 also activated the NRF2 reporter (heatmap depicting log2 fold change of either reporter vs. DMSO control in Fig. 2b), with papaverine and phenothiazine identified as the only exclusive activators of the GRHL reporter. The strongest activation of the GRHL reporter, considering all three tested concentrations for each compound, was observed with the compounds 2′,4′-Dihydroxy-4-methoxychalcone (DHM-chalcone, 286.2 ± 36.6 SEM at 20 μM, *P* = 0.008), alantolactone (131.5 ± 6.8 SEM at 20 μM, *P* = 0.017), and vorinostat (114.9 ± 6.0 SEM at 5 μM, *P* = 0.001). Notably, the HDACi vorinostat, while having the lowest efficacy of the top three GRHL reporter activators and in particular compared to DHM-chalcone, was most consistent in activating GRHL<sup>WT-LUC</sup> across the tested

range of concentrations, i.e., was observed to have the highest potency.

**GRHL activators extend nematodal lifespan dependent on GRH-1.** To explore whether GRHL reporter activation is indicative of compounds having longevity-promoting effects, all 15 confirmed primary hits were tested for their effect on lifespan of WT nematodes at a concentration of 1 μM (Fig. 2c–q). Overall, 9/15 compounds did not significantly alter lifespan at the tested concentration, and one compound (thimerosal, mean −33.4%, *P* < 0.001) shortened lifespan (Fig. 3a). The remaining compounds, including DHM-chalcone and alantolactone, significantly extended lifespan by +8.5–12.3% (Fig. 2c–g). The strongest effects (lifespan-extension >10%) were observed for the HDACi vorinostat (mean + 12.3%, *P* < 0.001) and two phytochemicals, papaverine (mean + 11.4%, *P* = 0.004) and piperlongumine (mean + 10.7%, *P* = 0.021). Both vorinostat and papaverine are compounds that are FDA-approved for pharmacological use. As the GRHL promoter element used here can be activated not only by GRHL1, but also by GRHL2 and GRHL3[16], specificity of vorinostat, papaverine, and piperlongumine in activating the reporter via GRHL1 was tested. To do so, using CRISPR/Cas9-mediated deletion, two independent GRHL reporter lines deficient for *GRHL1* but nor *2* or *3* were generated (GRHL<sup>NULL-LUC</sup>; Supplementary Fig. 3d, e). Relative average activation of GRHL<sup>NULL-LUC</sup> by vorinostat, papaverine, and piperlongumine, compared to activation of GRHL<sup>WT-LUC</sup> without *GRHL1* deletion (Fig. 3b), was significantly lower for all three compounds (vorinostat: 0.40 ± 0.05, $P = 3.9 \times 10^{-6}$; papaverine: 0.43 ± 0.05, $P = 1.2 \times 10^{-6}$; piperlongumine: 0.81 ± 0.05, *P* = 0.011), indicating that all partially act via GRHL1 to activate the human GRHL reporter.

Vorinostat, papaverine, and piperlongumine were each tested at two additional concentrations (0.1 and 10 μM) for their effect on *C. elegans* lifespan (Fig. 3c, d, f, g, i, j). Vorinostat and papaverine had

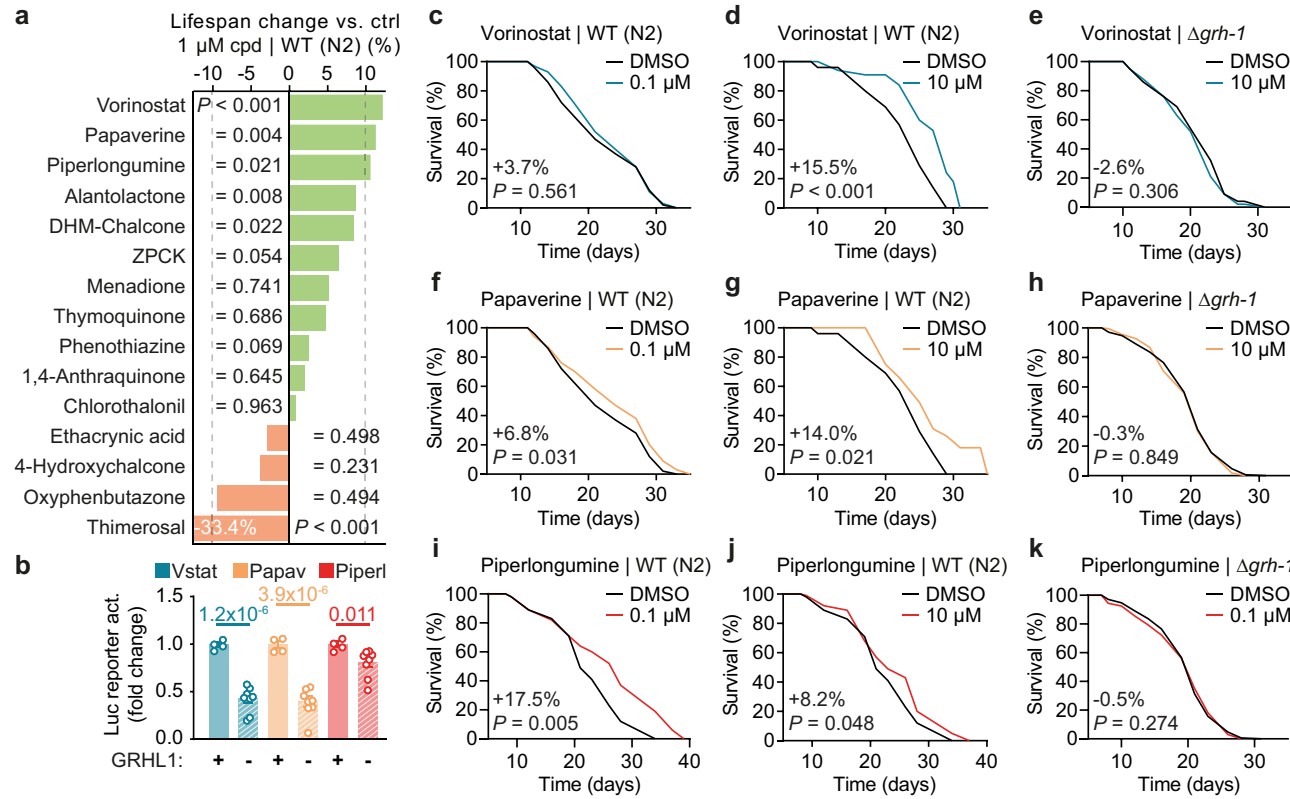

**Fig. 3 GRH-1 dependency of nematodal lifespan-extension elicited by selected GRHL1-activating compounds. a** Overview of impact of the respective compound (cpd) on lifespan of wild-type (WT) N2 nematodes at 1 µM in plate, as percentage change of mean lifespan vs. control treatment with DMSO (ctrl). Also see Fig. 2c–q. **b** Activation of GRHL reporter by the top 3 lifespan-extending compounds from (**a**) (Vstat = vorinostat, Papav = papaverine, Piperl = piperlongumine) at 5–10 µM in the presence (wild-type reporter with $n = 4$) or absence (two independent deletion reporters with $n = 8$ combined) of GRHL1. **c, d** Lifespan assays of vorinostat at 0.1 (**c**) or 10 µM (**d**) in plate, using WT (N2) nematodes and vs. DMSO control. **e** Lifespan assay of vorinostat at 10 µM in plate, using $\Delta grh$-1 and vs. DMSO control. **f, g** Similar to (**c**), (**d**), using papaverine. **h** Similar to (**e**), using papaverine at 10 µM in plate. **i, j** Similar to (**c**), (**d**), using piperlongumine. **k** Similar to (**e**), using piperlongumine at 0.1 µM in plate. Data in bar graphs are mean ± SEM, with sample sizes as stated and individual data points representing biological replicates. $P$-values were determined with two-tailed unequal variances $t$-tests of the indicated comparisons of unpaired control vs. treatment groups. Lifespan graphs contain percentage change of mean lifespan compound vs. DMSO control and $P$-values as determined by log-rank test. See Supplementary Data 1 for detailed lifespan assay statistics. Source data are provided as a Source data file.

the most pronounced lifespan-extending effect at 10 µM (vorinostat: mean +15.5%, $P < 0.001$; papaverine: mean +14.0%, $P = 0.021$, Fig. 3d, g), whereas piperlongumine was identified to be most effective at 0.1 µM (mean + 17.5%, $P = 0.005$; Fig. 3i). When applying these concentrations of the respective compound to $\Delta grh$-1, no changes in lifespan were observed anymore (Fig. 3e, h, k). This indicates that the tested compounds function through GRH-1 to exert their effects, and that this transcription factor is a mediator of pharmacologically induced longevity in nematodes. For subsequent experiments, we focused on the phytochemical papaverine, identified as the only lifespan-extending compound specific for activation of the GRHL reporter.

**Analysis of the human GRHL1 protein interactome.** To investigate how papaverine might impact GRHL1 activity, a HEK293-derived cell line overexpressing C-terminally 3xFLAG-6xHis tagged wild-type human GRHL1 (GRHL1[WT-TAG]) was generated (Supplementary Fig. 3f, g). GRHL1[WT-TAG] was verified to be located in the nucleus (Fig. 4a and Supplementary Fig. 3h) and as suitable for tandem-affinity purification (Supplementary Fig. 3i). The GRHL1[WT-TAG]-overexpressing cell line, together with HEK293 WT cells as background control, was subjected to DMSO vs. papaverine treatment for 24 h, nuclei isolated, GRHL1[WT-TAG] purified, and GRHL's post-translational modifications (PTMs) and its protein interactome determined by

liquid chromatography-mass spectrometry analysis (LC-MS; Fig. 4b, Supplementary Table 1 and Supplementary Data 5).

The most abundantly purified protein in the GRHL1[WT-TAG] samples reassuringly was GRHL1 itself. In addition, 130 additional proteins were identified as potential GRHL1[WT-TAG] interaction partners, i.e., fully absent or enriched ≥3-fold over the respective HEK293 WT control samples. Of these 130 proteins, 66 (50.8%) co-purified with GRHL1[WT-TAG] both after papaverine and DMSO control treatment, whereas 29 proteins (22.3%) were identified as potential GRHL1 interaction partners only following papaverine and 35 (26.9%) only following DMSO control treatment (Supplementary Data 5). All proteins were curated, and each assigned a category and classification reflecting their most prominent cellular function (Supplementary Fig. 3j and Supplementary Data 5).

The largest set of potential GRHL1 interaction partners as here identified is involved in transcriptional regulation (57 proteins), either as TFs themselves or as transcriptional coregulators. 19 of the potential GRHL1-interacting proteins are involved in chromatin organization and histone modification, consistent with a recently proposed role of human GRHL1-3 in regulating chromatin accessibility[26], notably by mechanisms and interactors not yet elucidated in detail. The only two experimentally identified human GRHL1 interactors described so far, in a large-scale study of the human protein–protein interactome[27], are GRHL2 and ZNF76. Our

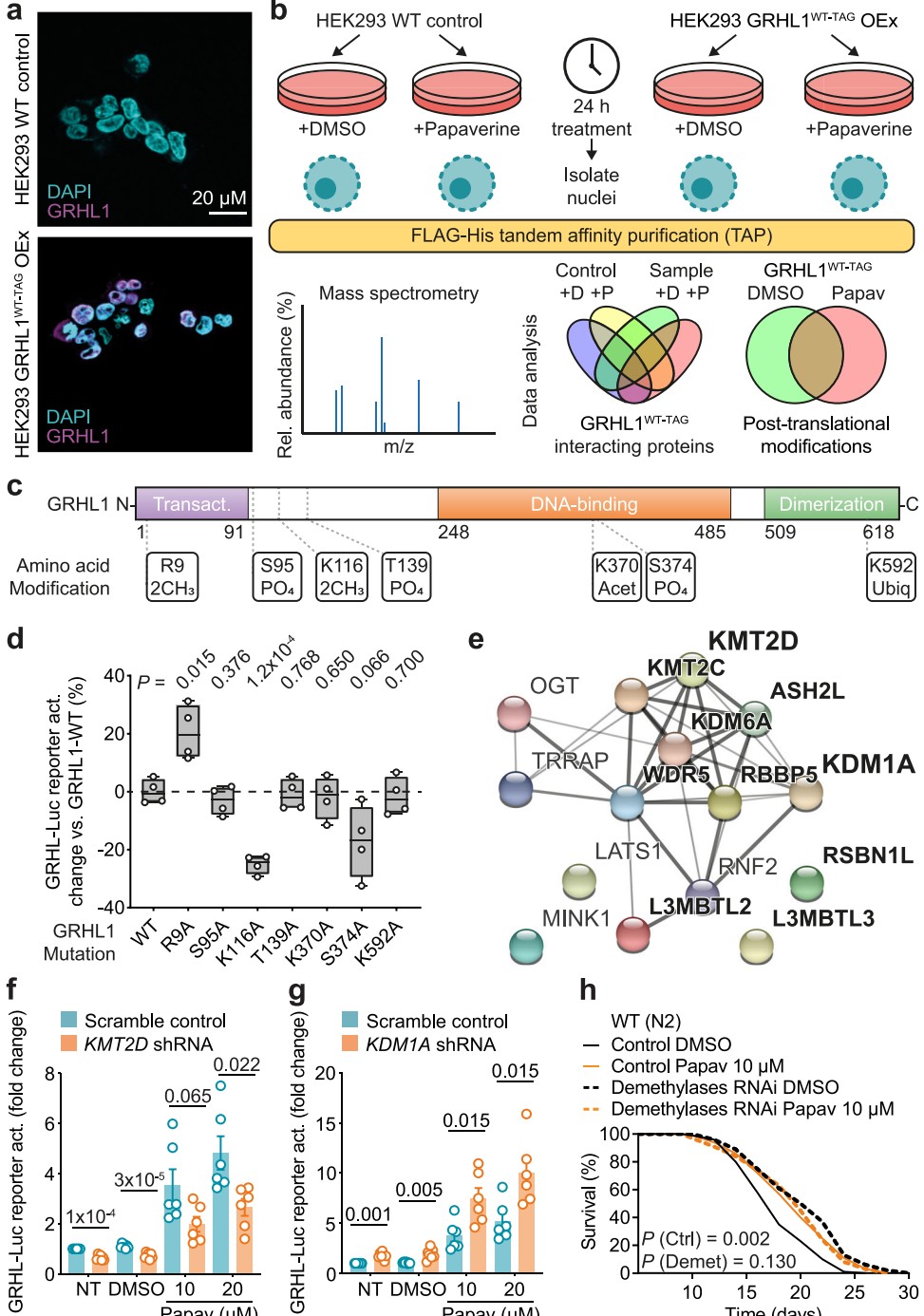

**Fig. 4 GRH-1/GRHL1 are regulated by lysine methylation. a** Microscopy images of a HEK293 wild-type (WT) and the GRHL1^WT-TAG-overexpressing cell line as DAPI staining and GRHL1 immunofluorescence overlay. See Supplementary Fig. 3h. **b** Experimental design of GRHL1^WT-TAG tandem-affinity purification (TAP) from isolated nuclei and subsequent mass spectrometry. **c** Post-translational modifications (PTMs) of human GRHL1 identified only after treatment with papaverine. Position, amino acid, and type of PTM as indicated (2CH$_3$ = dimethylation, PO$_4$ = phosphorylation, Acet = acetylation, Ubiq = ubiquitylation). See Supplementary Table 1 and Supplementary Data 6. **d** Impact of mutating individual GRHL1 amino acids on GRHL reporter activation. Depicted is percentage change vs. activation with the GRHL1 WT control with $n = 4$ for all. Box plot center lines indicate the median, bounds of boxes indicate the 25th to 75th percentiles, and whiskers indicate minima and maxima. **e** Protein–protein interaction network of GRHL1 interactors classified as impacting PTMs. Edges represent known protein–protein interactions, with darker color indicating stronger data support. Proteins in bold are linked to K-methylation. Visualized with the help of STRING (https://string-db.org). See Supplementary Fig. 3j and Supplementary Data 5. **f, g** Activation of the GRHL reporter without (scramble control) or with concomitant knockdown of *KMT2D* (**f**) or *KDM1A* (**g**) by shRNA and indicated treatments (NT = no treatment, DMSO = vehicle control, Papav = papaverine at 10 or 20 μM). *KMT2D* and *KDM1A* scramble vs. shRNA with $n = 7$ for NT and DMSO and $n = 6$ for Papav. **h** Lifespan assay of WT (N2) nematodes on post-developmental *lsd-1 + spr-5* demethylases RNAi compared to control vector L4440, and both conditions additionally treated with 10 μM papaverine or DMSO control. Data in bar graphs are mean ± SEM, with sample sizes as stated and individual data points representing biological replicates. *P*-values were determined with two-tailed unequal variances *t*-tests of the indicated comparisons of unpaired control vs. treatment groups. *P*-values of *C. elegans* lifespan assays were determined by log-rank test. See Supplementary Data 1 for detailed lifespan assay statistics. Source data are provided as a Source data file.

analysis confirmed GRHL2 as a potential interaction partner of GRHL1. Noteworthy, the GRHL1-GRHL2 interaction was observed only after papaverine treatment, suggesting that this heterodimer might be relevant in mediating transcriptional activation of target genes. While our analysis did not identify ZNF76 as an interaction partner, 14 potential interactors are other zinc finger proteins (including ZNF24, ZNF148, ZNF281, ZNF384, ZNF516, and ZNF638), suggesting a physiologically relevant link of GRHL1 to this highly abundant group of proteins that impacts a variety of processes in health and disease[28].

**GRHL1 activity is regulated by lysine methylation**. In parallel to the above, a total of 24 amino acids of GRHL1$^{WT-TAG}$ were identified as post-translationally modified using stringent protein identification filters, with 7 thereby designated as modified only after papaverine treatment (Fig. 4c, Supplementary Table 1, and Supplementary Data 6). This suggests involvement of these PTMs in regulating GRHL1 activity, especially since no modifications were identified exclusively in DMSO control samples by this approach. Interestingly, monitoring the LC-MS traces of the identified peptides across all samples, an overall higher abundance of GRHL1 was observed in papaverine-treated samples. The modifications initially defined as papaverine-specific based on the peptide identification results could also be observed in DMSO control samples to some extent (Supplementary Data 7). These 7 GRHL1 PTMs, including lysine acetylations, arginine and lysine dimethylations, threonine and serine phosphorylations, and lysine ubiquitylation, were selected for follow-up experiments. Reassuringly, our approach also identified the only GRHL1 PTM already described in a large-scale study of the human phosphoproteome, i.e., phosphorylation of threonine 208[29], in both the papaverine-treated and DMSO control samples.

To test whether the 7 GRHL1 PTMs only stringently identified in papaverine-treated samples are indeed involved in regulating GRHL1 activity, corresponding amino acid codons in the GRHL1$^{WT-TAG}$ overexpression construct were individually mutated to alanine. Each resulting vector (overexpressing tagged GRHL1 with mutation R9A, S95A, K116A, T139A, K370A, S374A, or K592A, respectively) was transiently transfected into the GRHL$^{WT-LUC}$ reporter and its relative effect on reporter activation, compared to transient transfection with the GRHL1$^{WT-TAG}$ vector, determined. This revealed that mutations R9A and K116A significantly impacted ability of GRHL1 to activate GRHL$^{WT-LUC}$ (Fig. 4d), with both R9 and K116 identified as dimethylated (me2) after papaverine treatment (Fig. 4c). The GRHL1$^{R9A-TAG}$ mutant displayed increased activation of the reporter ($+20.5 \pm 4.7\%$, $P = 0.015$), whereas activation by GRHL1$^{K116A-TAG}$ was reduced ($-25.0 \pm 1.6\%$, $P = 1.1 \times 10^{-4}$). This indicates that blocking R9 dimethylation promotes GRHL1 activity, thereby implying that R9me2 in WT GRHL1 is a repressing methylation that counteracts GRHL1 activity. Opposingly, when blocking dimethylation of K116, GRHL1 activated the transcriptional reporter less, from which follows that K116me2 in WT GRHL1 is likely an activating methylation.

To investigate this regulation of GRHL1 in more detail, we next considered all those GRHL1-interacting proteins which were categorized to be involved in impacting PTMs (Fig. 4e, Supplementary Fig. 3j, and Supplementary Data 5). Of the 15 proteins in this category, 10 are linked to K-methylation, among them K-methylases (KMT2C and KMT2D) and K-demethylases (KDM1A and KDM6A), and notably none to R-methylation or demethylation. Relevantly, transcriptional activity of p53 is repressed by KDM1A, likely by removal of an activating dimethylation mark of a single lysine residue[30], similar to our findings that mutation of K116 in GRHL1 reduces transcriptional

activity. Furthermore, KMT2C and KMT2D were recently also described as GRHL2-interacting proteins[31].

Impact of K-methylases and -demethylases on GRHL1 activity was tested by stably transfecting the GRHL$^{WT-LUC}$ reporter with shRNAs against *KMT2D* or *KDM1A*. Downregulation of *KMT2D* reduced reporter activity, both in the basal state and after papaverine treatment (Fig. 4f). This result is congruent with the interpretation that dimethylation of GRHL1 K116 is an activating mark, which is impaired when *KMT2D* is knocked down. Conversely, and further supporting a role for K-methylation, impairment of *KDM1A* led to increased GRHL reporter activity, again both under basal and induced conditions (Fig. 4g). This suggests that KDM1A could be responsible for removing the activating dimethylation mark of GRHL1, hence GRHL1 activity being increased when *KDM1A* is downregulated. In *C. elegans*, simultaneous RNAi impairment of the demethylases *lsd-1* and *spr-5*, which together encode the two highest confidence nematodal orthologues of human KDM1A, increased lifespan per se. Importantly, this longevity was not further extendable by additional treatment with papaverine (Fig. 4h), indicating a drug-gene interaction of papaverine with lysine demethylases also in nematodes.

**Grainyhead TFs are linked to regulation of insulin signaling**. To further elucidate the mechanism of GRHL1 activation by papaverine, we performed pull-down experiments using de novo synthesized biotinylated papaverine (Supplementary Fig. 4a) added to HEK293 cell extracts (Supplementary Fig. 4b). Thereby, 25 human proteins were identified as potential papaverine interactors, co-purifying exclusively or enriched ≥3-fold with biotinylated papaverine over control. Notably, the majority of these proteins are linked to insulin/IGF-1 and/or the related PI3K-AKT-mTOR signaling axis (Supplementary Fig. 4c and Supplementary Table 2). Among the proteins co-purifying with papaverine, we decided to focus in detail on the phosphoinositide 3-kinase (PI3K) PIK3C2A. PI3Ks have been previously reported to oppose KMT2D[32] and to promote KDM1A[33], both here identified as GRHL1 interactors influencing activation mediated by papaverine. In addition, papaverine has been independently linked to downregulation of PI3K signaling[34]. We hence hypothesized that papaverine might affect PIK3C2A, which in turn could lead to activation of KMT2D and inhibition of KDM1A, thus altering activity of GRHL1.

Impairing *PIK3C2A* by shRNA in the GRHL$^{WT-LUC}$ reporter (Fig. 5a), equivalent to the experiments performed for *KDM1A* and *KMT2D*, led to increased GRHL reporter activation, overall resembling the pattern observed for *KDM1A* impairment (Fig. 4g). The closest orthologue of human PIK3C2A in nematodes is a PI3K named PIKI-1. Treatment of nematodes with RNAi against *piki-1* increased lifespan per se (Fig. 5b), similar to impairment of the nematodal *KDM1A* orthologues *lsd-1* and *spr-5* (Fig. 4h). Additional application of papaverine failed to further extend lifespan, again indicating epistatic interaction of this compound with PIKI-1/PIK3C2A (Fig. 5b). Together, these results are consistent with GRHL1 activity after papaverine treatment being promoted by K116 dimethylation and modulated by KMT2D, KDM1A, and PIK3C2A.

Given that PI3Ks are arguably best known as mediators of insulin/IGF-1 signaling[35], we tested whether GRH-1 might be involved in this pathway. *C. elegans* loss-of-function mutations of *daf-2* (encoding the insulin/IGF-1 receptor orthologue) and *age-1* (encoding the canonical PI3K orthologue) are well described to extend lifespan in a manner requiring DAF-16/FOXO[36–38], a downstream TF of IIS conserved from *C. elegans* to mammals[13]. In mRNA samples from both *daf-2* and *age-1* mutants, *grh-1*

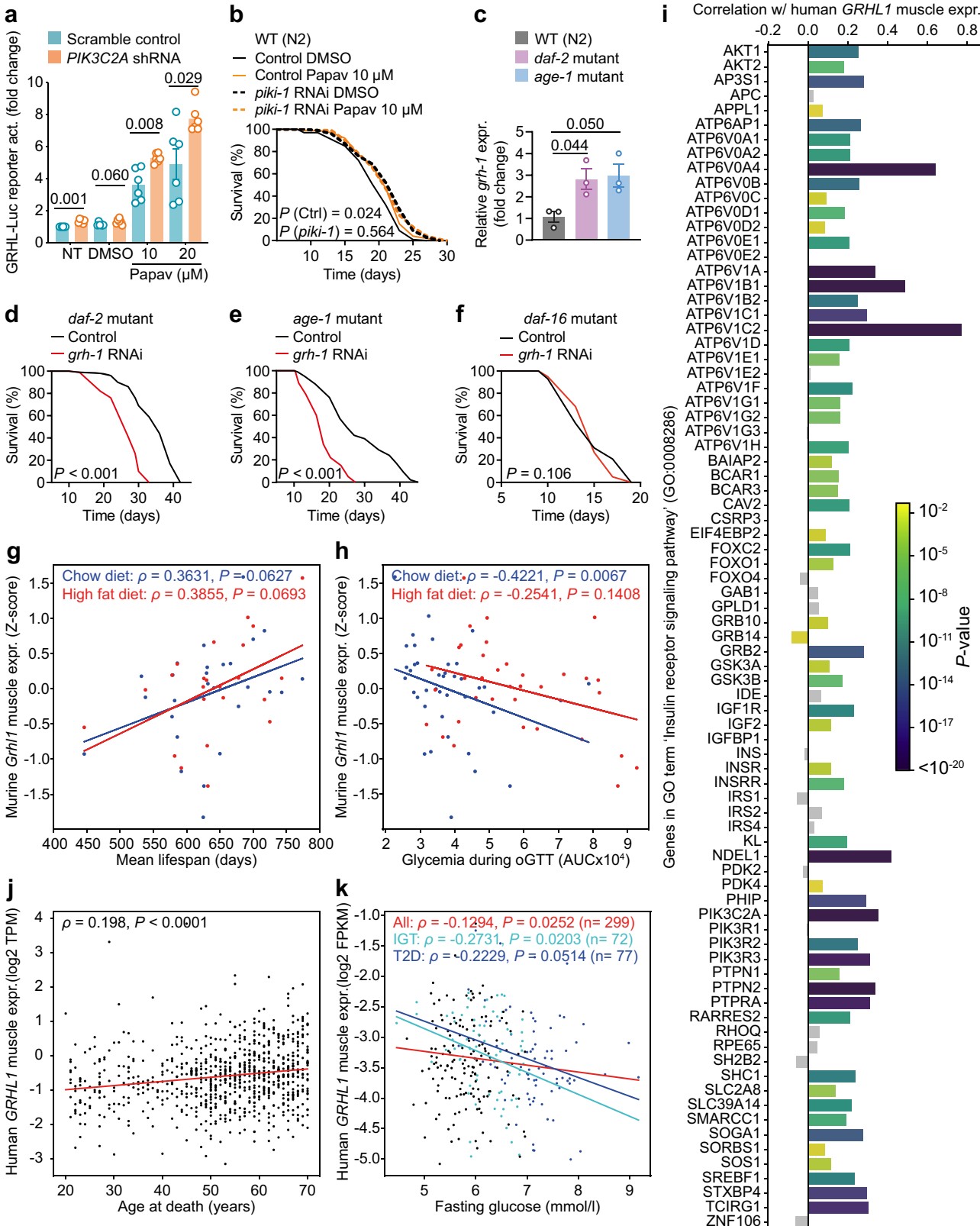

expression was upregulated compared to WT (Fig. 5c), and application of *grh-1* RNAi to either long-lived mutant markedly reduced their lifespan (Fig. 5d, e) to WT levels. Conversely, in the short-lived *daf-16* mutant strain, application of *grh-1* RNAi did not further reduce lifespan (Fig. 5f). This indicates a possible functional interaction between DAF-16/FOXO and GRH-1 as

downstream transcriptional mediators of IIS. Consistently, extended lifespan of *grh-1* OEx was fully abolished following impairment of *aak-2* (AMPK), *daf-16* (FOXO), or *skn-1* (NRF2) by RNAi (Supplementary Fig. 4d–g), i.e., components established as required for IIS-mediated longevity[38,39]. Moreover, *grh-1* overexpression was unable to further extend the longevity

**Fig. 5 Nematodal and mammalian Grainyhead TFs are linked to aging and insulin signaling. a** Activation of the GRHL reporter without (scramble control) or with concomitant knockdown of *PIK3C2A* by shRNA and indicated treatments (NT = no treatment, DMSO = vehicle control, Papav = papaverine at 10 or 20 µM; n = 6 for all). **b** Lifespan assay of wild-type (WT) N2 nematodes on post-developmental *piki-1* RNAi compared to control vector L4440. Both conditions additionally treated with 10 µM papaverine or DMSO control. **c** RT-qPCR of *grh-1*, in WT (N2) vs. a *daf-2* or *age-1* mutant strain (n = 3 for all). **d–f** Lifespan assays with a *daf-2* (**d**), *age-1* (**e**), or *daf-16* (**f**) mutant strain on post-developmental *grh-1* RNAi compared to control vector L4440. **g** Spearman's rho correlation between *Grhl1* expression in muscle (Z-score normalized) and mean lifespan of chow and high-fat diet-fed mice from the BXD mouse genetic reference population. **h** Similar as in (**g**), depicting the correlation of *Grhl1* expression in muscle with area under the curve (AUC) glycemia during oral glucose tolerance test (oGTT). **i** Pearson's *r* correlation coefficients between *GRHL1* and genes in GO term 'Insulin receptor signaling pathway' (GO: 0008286) in human muscle samples, derived from the GTEx database. Gray bars represent non-significant correlations (*P*-value > 0.05). **j** Spearman's rank rho correlation between *GRHL1* expression in human skeletal muscle and age at natural death from the GTEx v8 dataset (n = 621). **k** Spearman's rank rho correlation between *GRHL1* expression in human skeletal muscle and fasting blood glucose in the entire FUSION cohort (All) or in individuals with impaired glucose tolerance (IGT) or type 2 diabetes (T2D). Color of dots indicates IGT individuals (light blue), T2D individuals (blue), and individuals belonging to neither group (black). Data in bar graphs are mean ± SEM, with sample sizes as stated and individual data points representing biological replicates. *P*-values were determined with two-tailed unequal variances *t*-tests of the indicated comparisons of unpaired control vs. treatment groups. *P*-values of *C. elegans* lifespan assays were determined by log-rank test. See Supplementary Data 1 for detailed lifespan assay statistics. Source data are provided as a Source data file.

phenotype elicited by *daf-2* RNAi treatment (Supplementary Fig. 4h).

Having uncovered a role for nematodal GRH-1, aside from affecting life- and healthspan, in impacting IIS, we next investigated whether these roles might be evolutionarily conserved in mammals. Strikingly, in the BXD mouse genetic reference population[19] *Grhl1* expression in skeletal muscle of mice on both a chow and high-fat diet predicts mean lifespan by trend (Fig. 5g). In addition, *Grhl1* expression in skeletal muscle of animals on a chow diet negatively correlates with glycemia during oral glucose tolerance test (oGTT) (Fig. 5h), indicating increased insulin sensitivity. When considering murine *Grhl1* expression in liver, a positive correlation with mean lifespan was again observed for animals fed with a chow diet, while such an effect was not detected for animals on a high-fat diet (Supplementary Fig. 5a). On neither diet a correlation of *Grhl1* liver expression with glycemia during oGTT was found (Supplementary Fig. 5b).

Focusing lastly on human *GRHL1*, gene set enrichment analysis revealed its expression to consistently positively correlate with genes present in pathways related to insulin signaling across several tissues (Supplementary Fig. 5c). In particular, the majority of genes contained within the GO term 'insulin receptor signaling pathway' (GO:0008286), including *PIK3C2A*, are positively correlated with human *GRHL1* expression in skeletal muscle (Fig. 5i), and similar correlations were observed with *GRHL1* expression in liver (Supplementary Fig. 5d).

To further investigate whether human *GRHL1* expression, following the observations in BXD mice, also correlates with lifespan and glucose tolerance, we analyzed the GTEx human cohort[40]. Thereby, we discovered that *GRHL1* expression in skeletal muscle positively correlates with age at death (Fig. 5j and Supplementary Fig. 5e). Next, we analyzed the FUSION tissue biopsy study dataset[41,42], which includes tissue biopsy gene expression data and the glucose tolerance status of participants (Supplementary Fig. 5f). Human skeletal muscle *GRHL1* expression negatively correlates with fasting blood glucose not only when considering every participant of the cohort, but even more evidently so in individuals with impaired glucose tolerance (IGT), or type 2 diabetes (T2D) (Fig. 5k). These data suggest that GRHL1 is a critical regulator of glucose homeostasis also in humans. Taken together, these observations underscore a link of Grainyhead 1 both to regulation of lifespan and insulin signaling that is conserved from nematodes to mammals.

## Discussion

Aging is arguably the most pervasive and simultaneously least understood biological phenomenon. Nevertheless, research over

the last decades showed that aging (i) is influenced by environmental and genetic factors[1,2]; (ii) is a unifying risk factor for human diseases[5]; and (iii) can be impacted by various interventions across species[3,6,7]. The development of translational medicine applications to address human aging-associated diseases, and potentially to slow aging itself, requires identification of cellular targets relevant for controlling aging and externally modulable, e.g., by pharmacological treatment[3,6,7]. Here, we identified and characterized an evolutionarily conserved longevity-promoting transcription factor as a target for geroprotective interventions, further supporting the role transcriptional regulation is recognized to play in the aging process[8,10–12].

Based on mammalian TF taxonomy, the specific protein belongs to the immunoglobulin fold superclass, and therein to the class of Grainyhead domain factors, and is termed Grhl1/GRHL1 in humans and mice, respectively, and GRH-1 in *C. elegans*. To date, Grainyhead TFs are mainly investigated regarding their role during murine development[43,44], and for their complex influence on development and progression of various cancers[45]. More recently, a functional role for Grainyhead TFs in regulating the chromatin landscape and mediating altered epigenetic states has been described[26]. We now find that GRH-1 is also acting as an anti-aging TF in a dose-dependent manner. While the underlying mechanisms for this dose-dependency are subject to clarification, we believe this to be an important observation that highlights necessary considerations when aiming to manipulate TFs to counteract aging and in the context of addressing aging-associated diseases. As exemplified by NRF2's dual role as both a potential tumor suppressor and oncogenic protein in various cancers[46], the degree to which a particular aging- or disease-related TF is activated or inhibited, in addition to often observed tissue- and context-dependency of effects, needs to be considered when designing appropriate interventions. We further identified GRH-1 as involved in regulating MAPK signaling, a role likely conserved for human GRHL1, and to improve nematodal survival upon exposure to bacterial pathogens. Given the known role of Grainyhead TFs in regulating epithelial barrier formation across species, including in mammals[44], it appears likely that mammalian Grhl1/GRHL1 might have comparable functions in opposing bacterial infections, which presents a relevant focus for future investigation. Our findings thus add to the already manifold functional roles of Grainyhead TFs, thereby highlighting the versatility of key transcriptional regulators in impacting diverse molecular, cellular, and physiological processes.

As proof-of-principle for developing pharmacological interventions targeting Grainyhead TFs with geroprotective drugs, we here established a human reporter cell line to identify small-

molecule GRHL1 activators that also promote nematodal longevity via GRH-1. Among the strongest GRH-1-dependent lifespan-extending compounds here discovered was vorinostat, functionally classified as an HDAC-inhibitor. Interestingly, several studies have independently identified HDACis as candidate geroprotective compounds in nematodes, flies, mice, and human in vitro models[47]. With the underlying mechanisms still incompletely understood and actively investigated, we here provide evidence that individual TFs, such as GRH-1/Grhl1/GRHL1, might be responsible for mediating beneficial effects associated with HDACi-treatment. Piperlongumine, a plant alkaloid also exhibiting HDAC-inhibiting functions[48], was likewise found here to extend nematodal lifespan dependent on GRH-1. Of note, piperlongumine has been described to act as a senolytic agent by specifically promoting cell death of senescent cultured fibroblasts[49]. The evidence we provide support it as a lead compound for developing pharmacological anti-aging treatments and overall indicate that utilization of in vitro models and comparatively simple non-vertebrate animal model organisms is a valid experimental strategy to identify candidate geroprotectors.

Among the human GRHL1-activating compounds extending nematodal lifespan, we specifically focused on papaverine as a representative activator, the only compound in our screen that failed to also activate NRF2. Papaverine is mainly known for its longstanding medical use as a vasodilator and its function as a non-selective phosphodiesterase inhibitor[50], and to our knowledge, unlike vorinostat and piperlongumine, is hitherto not linked to regulation of lifespan or aging-related processes. We found treatment with papaverine to impact the human GRHL1 protein–protein interactome and several treatment-dependent potential interactors to be linked to aging and age-related pathologies. Examples are the TF MYC, a proto-oncogenic protein[51] and core member of the MYC-MAX-MAD network, heterozygous deletion of which increases murine lifespan[52]; the RecQ helicase WRN, whose loss-of-function mutation causes the progeroid Werner syndrome in humans[53,54]; the two AAA+ ATPases RUVBL1/PONTIN and RUVBL2/REPTIN, telomerase components essential for holoenzyme assembly[55], with improper telomere maintenance being widely implicated in aging and disease[56]. Notably, this is the first comprehensive description of the human GRHL1 protein–protein interactome, which overall appears to support a role for Grainyhead TFs in the aging process. It should be noted, however, that further analyses of individual potential GRHL1-interacting proteins here identified, e.g., by performing reverse affinity purification experiments of candidates, are required to validate their physical interaction with GRHL1. Probing in detail the observed potential GRHL1 interaction partners in future studies thus might lead to confirmation of as yet unrecognized interconnections between different aging-related pathways.

Focusing for now on the regulation of GRHL1 itself, we have found its activity to be likely impacted by post-translational dimethylation of a specific lysine residue (K116) and to be opposingly modulated by a K-methylase (KMT2D) and K-demethylase (KDM1A aka LSD1), both here identified as potential GRHL1-interacting proteins. We furthermore observed nematodal lifespan-extension by papaverine to be impacted by RNAi of K-demethylase orthologues (lsd-1 and spr-5), thus indicating a potential mechanistic conservation of this signaling axis. Nevertheless, it currently remains unclear whether nematodal GRH-1 is indeed subject to similar post-translational modifications as GRHL1, in particular since K116 and also other amino acids found modified are not conserved between the C. elegans and human protein. Notably, our findings for human GRHL1 are consistent with a similar lysine-dependent regulation of p53 activity, also involving interaction with KDM1A/LSD1[30], with p53 structurally related to Grainyhead TFs

and part of the same immunoglobulin fold superclass[16]. In addition, K-methylases were recently also described as interaction partners of GRHL2[31]. This property is possibly related to the role of Grainyhead TFs in regulating the chromatin landscape by recruiting epigenetic modifiers[26], for which our description of the GRHL1 protein–protein interactome provides a first comprehensive resource for studies in this regard.

By performing a protein pull-down using biotinylated papaverine, to further clarify the pathway leading to GRHL1 activation, we identified the PI3 kinase PIK3C2A as a potential papaverine interaction partner. It is important to note that in chemical proteomics, immobilization might alter activity and/or change the interaction profile of a lead compound depending on the attachment site. Therefore, we here used a differential proteomics approach to compare the relative abundance of proteins purified together with the probe (i.e., biotinylated papaverine) vs. those purified together with the biotinylated linker control, assessing only proteins that are enriched ≥3-fold as potential binding partners. As the control consists of the full structure up to the pharmacophore (i.e., papaverine), all differential binding partners are likely due to protein-compound interactions through the pharmacophore. Whereas it cannot be ruled out that certain binding partners might be missed due to the nature of the attachment site, the method here used enables identification of specific targets and avoids nonspecific binders. Notably, others independently found papaverine as linked to regulation of PI3K signaling[34] and PI3Ks are previously reported to modulate the activity of both KMT2D[32] and KDM1A[33]. Together with our findings, this suggests that papaverine might first act on PIK3C2A, which then in turn regulates KMT2D and KDM1A to alter the K-methylation state and thereby activity of GRHL1. Interestingly, GRHL1 and biotinylated papaverine share the probable serine carboxypeptidase CPVL (Swiss-Prot ID: Q9H3G5), an as yet only superficially characterized protein, as their only common potential interaction partner. While we have not followed up further on this finding in the context of the present study, it is a quite exciting prospect for future research.

Presently, we were particularly curious of the connection of PI3Ks to insulin signaling[18], modulation of which is prominently linked to regulation of longevity across species[36,37,57], and whether Grainyhead TFs might also be involved in this fundamental metabolic pathway. Indeed, we found grh-1 expression in nematodes to be increased in two long-lived loss-of-function mutants of the central IIS pathway components daf-2 (encoding the nematodal insulin/IGF-1 receptor orthologue) and age-1 (encoding the canonical IIS PI3K orthologue). Furthermore, GRH-1 is required to maintain the lifespan-extension of daf-2 and age-1 mutant strains, reminiscent of the transcription factor DAF-16/FOXO[37,38].

Supporting an evolutionarily conserved role of Grainyhead TFs in aging and IIS, we found that in the BXD mouse genetic reference population[19] Grhl1 expression especially in muscle of mice, and consistently so for animals undergoing different dietary interventions, not only positively correlates with an increased mean lifespan, but at the same time also improves insulin sensitivity, as evidenced by a negative correlation with blood glucose levels in an oral glucose tolerance test. Moreover, we found human GRHL1 expression to be consistently positively correlated with insulin signaling across tissues, and in particular in both muscle and liver to be strongly genetically correlated with the majority of insulin receptor signaling pathway components. Finally, highly reminiscent of the observations in BXD mice and thereby suggesting functional conservation of Grhl1/GRHL1 in mammals, we found that human skeletal muscle GRHL1 expression, in the GTEx cohort[40], positively correlates with increased age at death and, in the FUSION tissue biopsy study

dataset[41,42], negatively with fasting blood glucose levels, including in individuals with impaired glucose tolerance or type 2 diabetes. Notably and in general, when attempting to interpret and potentially translate findings regarding the impact of IIS modulation in different species, similarities and discrepancies between, e.g., *C. elegans* and mammals need to be carefully considered. While *daf-2* impairment by RNAi and *daf-2* incomplete loss-of-function mutations both extend nematodal lifespan, partial impairment of the insulin receptor in mice does not[37,58], and impaired insulin signaling in humans is linked to the development of metabolic syndrome[59]. By contrast, indicating a certain degree of evolutionary conservation, the glucometabolic effect of *daf-2* impairment is very similar to that of impaired insulin signaling in mammalian cells, in both cases leading to impaired glucose uptake and transport[60]. Thus, while our findings clearly indicate a functional connection of Grainyhead TFs to insulin signaling in different species, including in humans, delineating their relevance for this central metabolic pathway in mammals will require detailed investigations and the use of more sophisticated models, e.g., to selectively impair or overexpress murine *Grhl1* in liver or other organs and tissues important for IIS.

In summary, our results are consistent with Grainyhead TFs, in particular GRH-1/Grhl1/GRHL1, having an evolutionarily conserved role in regulating lifespan and insulin signaling in nematodes and mammals. They further identify these transcription factors as potential targets for the development of geroprotective drugs, for which small-molecule compounds such as papaverine, piperlongumine, and vorinostat might already serve as leads. Applying similar screening strategies, as here utilized for proof-of-principle and basic research purposes, to more extensive compound libraries can be expected to facilitate discovery of additional novel drug candidates in the quest to ameliorate metabolic diseases and slow human aging.

## Methods

**C. elegans strains.** The following *C. elegans* strains used for this publication were provided by the *Caenorhabditis* Genetics Center (CGC, University of Minnesota, USA): Wild-type N2 (Bristol), RB754 *aak-2* (ok524), TJ1052 *age-1* (hx546), CB1370 *daf-2* (e1370), CF1038 *daf-16* (mu86), VC2072 *grh-1* (gk960), RB1813 *piki-1* (ok2346), EU31 *skn-1* (zu135). Strains overexpressing *grh-1* were generated as detailed below and are termed MIR260 (risIs31) (with *hsp-16.2*-promoter and referred to as *grh-1* OEx line 1 in the main text), MIR261 (risIs31) (with *hsp-16.2*-promoter and referred to as *grh-1* OEx line 2 in the main text), and MIR262 (risIs32) (with endogenous *grh-1* promoter). For maintenance, nematodes were grown on Nematode Growth Medium (NGM) agar plates in 90 mm petri dishes at 20 °C using *E. coli* OP50 bacteria as a food source[61]. For experiments involving transient heat shock, nematodes on first day of adulthood were transferred to 33 °C for 1 h and then returned to 20 °C, with the respective wild-type control nematodes subjected to the same procedure. NGM agar plates, after pouring, were dried at room temperature for 1–2 days and then stored at 4 °C until further use.

**Bacterial strains.** *E. coli* OP50 bacteria (CGC) were streaked out on DYT agar plates and single colonies picked from such plates were cultured overnight at 37 °C and constant shaking in Erlenmeyer flasks containing liquid DYT medium. Bacterial overnight cultures were concentrated by centrifugation for 30 min at 3200 × g and 4 °C. The prepared bacteria were spotted on NGM agar plates and allowed to grow for 16–24 h prior to use.

*E. coli* HT115(DE3) RNAi bacteria (CGC) were streaked out on LB agar plates with 100 µg/ml Ampicillin and 12.5 µg/ml Tetracycline and single colonies picked from such plates were cultured overnight at 37 °C and constant shaking in Erlenmeyer flasks containing liquid LB medium with 100 µg/ml Ampicillin. Bacterial overnight cultures were concentrated by centrifugation for 30 min at 3200 × g and 4 °C. The prepared bacteria were spotted on NGM agar plates additionally containing 100 µg/ml Ampicillin and 1 mM Isopropyl-beta-D-thiogalactopyranoside (IPTG) and allowed to grow for 16–24 h prior to use (reagents from AppliChem).

For preparation of heat-inactivated (HIT) OP50, the overnight culture was pelleted by centrifugation as above, all DYT media removed, and bacteria resuspended in S-buffer supplemented with 1 M MgSO₄ and 5 mg/ml Cholesterol to a 10-fold concentrated culture. Afterward, the bacterial suspension was placed in

a 65 °C water bath for 45 min. HIT OP50 were spotted on NGM agar plates on the day of use and dried for 30 min before adding the worms.

Pathogenic bacterial strains used in this study were *Microbacterium nematophilum* (CGC, #CBX102) and *Pectobacterium carotovorum* (BCCM, #LMG2404). *M. nematophilum* was grown for 48 h in liquid LB medium at 37 °C and diluted 10-fold with OP50 or HT115 bacteria before being spotted on NGM agar plates. *P. carotovorum* was grown overnight at 37 °C in DYT medium at 37 °C and spotted on NGM agar plates pure.

**Human cell culture.** HEK293 cells were obtained from a commercial source (CLS Cell Lines Service GmbH, #300192) and cultured in DMEM (Bioswisstec AG, #M 1434) with pH 7.4 and 4.5 g/l glucose, containing phenol red, 10% FBS, and 1% penicillin/streptomycin at 37 °C, 5% CO₂, and 95% relative humidity. The different stable transgenic HEK293 cell lines used in this study are all derived from this initial cell line and were cultured under the same conditions, with the following antibiotics added: GRHL and ARE/NRF2 luciferase reporter and *GRHL1* overexpressing cell lines +100 µg/ml hygromycin B; GRHL luciferase reporter with concomitant *GRHL1* deletion +100 µg/ml hygromycin B and +0.25 µg/ml puromycin. For subculturing, HEK293 cells were commonly split at a 1:4–1:8 ratio every 2–3 days. Mouse embryonic fibroblasts (MEFs) derived from a conditional *Grhl1* knockout mouse model were cultured using DMEM without phenol red but otherwise using the same general conditions as for HEK293 cells, with a split ratio of 1:2–1:4 every 3–4 days, and all MEF experiments were performed at a total passage number lower than five. Generation of the different cell lines and 4-hydroxytamoxifen induction of MEFs are detailed below.

**Molecular cloning procedures.** Vectors to overexpress *grh-1* were generated with MultiSite Gateway cloning using the Gateway BP (#11789020) and LR (#11791020) Clonase II Enzyme mix according to the manufacturer's instructions. The *grh-1* full-length ORF without stop codons and with flanking attB1 and attB2 recombination sites was amplified from *C. elegans* cDNA using primers attB1_grh-1_fwd and attB2_grh-1_rev, with the amplified isoform identified as *grh-1b* by sequencing, and inserted into vector pDONR 221 (#12536017, all reagents from Thermo Fisher Scientific). A 1951 bp fragment of the endogenous *grh-1* promoter with flanking attB4 and attB1R recombination sites was amplified from *C. elegans* gDNA, using primers attB4_grh-1p_f and attB1R_grh-1p_r. Similarly, the 348 bp *hsp-16.2* promoter was amplified with primers attB4_hsp16.2p_fwd and attb1R_hsp16.2p_rev. Both promoters were individually inserted into pDONR P4-P1R. The different promoter-ORF combinations were inserted into the destination vector pdestMB14 (Addgene, #26415) in-frame with the sequence encoding a C-terminal GFP. The resulting vectors (pdestMB14_hsp-16.2p-grh-1-gfp, Addgene, #177762, and pdestMB14_grh-1p-grh-1-gfp, Addgene, #177761) were used to generate corresponding *C. elegans* *grh-1* overexpressing strains: pdestMB14_hsp-16.2p-grh-1-gfp to generate MIR260 (risIs31) and MIR261 (risIs31), and pdestMB14_grh-1p-grh-1-gfp to generate MIR262 (risIs32).

Luciferase reporter vectors for GRHL and ARE/NRF2 are based on the vector pGL4.27 (Promega Corporation, #E8451). Synthetic promoter elements (SPE) with several repeats of the mammalian GRHL1-3 consensus binding sequence[16] or the NRF2-bound antioxidant response element (ARE)[24], flanked by 5′-KpnI and 3′-NheI restriction sites, were obtained by custom DNA synthesis and inserted in front of the pGL4.27 minimal promoter using standard restriction-ligation procedures. The resulting vectors (pGL4.27_GRHL-SPE, Addgene, #177774, and pGL4.27_ARE/NRF2-SPE, Addgene,#177775) were used to generate corresponding stable HEK293 luciferase reporter cell lines.

The vector to overexpress the full-length human *GRHL1* ORF is based on vector pcDNA 3.1/Hygro(+) (Thermo Fisher Scientific, #V87020), which was first modified by inserting a sequence encoding a linker followed by a 3xFLAG-6xHis tag (L3F6H), using KpnI and BamHI restriction sites. The *GRHL1* ORF without stop codon and with flanking NheI and KpnI restriction sites was amplified from HEK293 cDNA using primers GRHL1_NheI_fwd and GRHL1_KpnI_rev and inserted into pcDNA 3.1/Hygro(+)-L3F6H using standard restriction-ligation procedures. The resulting vector (pcDNA3.1_Hygro(+)_GRHL1-L3F6H, Addgene, #177765) was used to generate a HEK293 cell line stably overexpressing C-terminally 3xFLAG-6xHis tagged wild-type human GRHL1. For transient transfection of wild-type or mutated *GRHL1* into the GRHL luciferase reporter cell line, individual amino acid codons in pcDNA3.1_Hygro(+)_GRHL1-L3F6H were mutated with the Q5 Site-Directed Mutagenesis Kit (New England Biolabs, #E0554S) and primers selected with https://nebasechanger.neb.com/. The seven resulting vectors are also deposited at Addgene (#177767-177773).

Vectors for CRISPR/Cas9-mediated deletion of 2860 bp of human *GRHL1* gDNA (partial exon 5, full exons 6 and 7, and partial exon 8) are based on vector pSpCas9(BB)-2A-Puro (PX459) V2.0 (Addgene, #62988) and were cloned according to the detailed protocol provided by the Zhang lab[62]. Top and bottom strand oligonucleotides with 20 bp *GRHL1*-specific guide sequences were annealed and inserted into pSpCas9(BB)-2A-Puro (PX459) V2.0 using scarless cloning with the BbsI restriction enzyme. Oligonucleotides were designed to target sequences followed by NGG at their 3′-end in the genomic context, located in exon 5 (5′-TCCGCAGACTGAGATCCGAG-3′) and exon 8 (5′-CAACACTATCAGTAA CATCG-3′), respectively. The resulting vectors (PX459_GRHL1-exon5, Addgene, #177763, and PX459_GRHL1-exon8, Addgene, #177764) were co-transfected into

a HEK293 GRHL luciferase reporter cell line to obtain the respective reporter with a concomitant *GRHL1* loss-of-function mutation. Sequences are listed in Supplementary Table 3.

**Bombardment and generation of stable *C. elegans* strains**. The pdestMB14 vectors for overexpression of *grh-1* under control of the endogenous *grh-1* promoter or the *hsp-16.2* promoter were transformed into the *unc-119*-deficient *C. elegans* strain HT1593 (*unc-119*(*ed3*) III) by microparticle bombardment using the biolistic particle delivery system PDS-1000/He (Bio-Rad) according to the manufacturer's instructions and previously described protocols[63], with in total 7 µg of vector DNA per bombardment. Homozygous transgenic lines with stable integration of the respective vector (MIR260 (risIs31) and MIR261 (risIs31) with pdestMB14_hsp-16.2p-grh-1-gfp, and MIR262 (risIs32) with pdestMB14_grh-1p-grh-1-gfp) were selected based on GFP fluorescence (for *hsp-16.2* promoter constructs, selection by GFP was performed after transient 1 h heat shock at 33 °C) and backcrossed at least four times against wild-type N2 nematodes before being used in any experiments.

**Lipofection and generation of stable human cell lines**. All transient and stable transfections of HEK293 cells were carried out using the Lipofectamine 3000 Transfection Reagent (Thermo Fisher Scientific, #L3000008) according to the manufacturer's instructions. To obtain stable transgenic cell lines, lipofection was performed in 6-well plates and cells were transferred to 15 cm dishes after 24 h. Selection with the appropriate antibiotics was started after 48 h (hygromycin B at 200–300 µg/ml and puromycin at 0.5–1.0 µg/ml). After 7–12 days of selection, individual clones were picked using cloning discs and expanded for testing. Luciferase reporter cell lines were screened by performing a luciferase assay using appropriate positive control compounds, and *GRHL1* overexpressing cell lines checked by western blot (see below). Homozygous partial deletion of *GRHL1* in the GRHL luciferase reporter was confirmed by genotyping PCR using primers GRHL1_del_fwd, GRHL1_del_rev, and GRHL1_wt_rev. For generation of GRHL luciferase reporters with shRNA-mediated individual knockdown of *KMT2D*, *KDM1A*, or *PIK3C2A*, the reporter was transfected with a respective commercial shRNA or scramble control vector (all vectors from OriGene, KMT2D: #TL311461, KDM1A: #TL316984, PIK3C2A: #TL310432) carrying a puromycin resistance. Knockdown was confirmed by RT-qPCR using primers KMT2D_qPCR_fwd + KMT2D_qPCR_rev, KDM1A_qPCR_fwd + KDM1A_qPCR_rev, and PIK3-C2A_qPCR_fwd + PIK3C2A_qPCR_rev, respectively. Monoclonal cell lines with the respective modifications were cultured for several passages and verified as stable before being used for the appropriate experiments. Sequences are listed in Supplementary Table 3.

**Compound library and compound preparation**. 'The Spectrum Collection' compound library (MicroSource Discovery Systems), was obtained at 10 mM in DMSO in 32 × 96-well plates with 80 compounds per plate. Upon arrival, plates were aliquoted and stored at −80 °C until use. Individual compounds were sourced from different suppliers (CAS/Product #). abcr GmbH: 1,4-Anthraquinone (635-12-1/AB177277). Santa Cruz Biotechnology, Inc: Scriptaid (287383-59-9/sc-202807). Selleck Chemicals LLC: Alantolactone (546-43-0/S8318); piperlongumine (20069-09-4/S7551); vorinostat (149647-78-9/S1047). Sigma-Aldrich/Merck KGaA: 2′,4′-Dihydroxy-4-methoxychalcone (81674-91-1/IDF00081); 4-hydroxychalcone (20426-12-4/S350664); chlorothalonil (1897-45-6/36791); DL-sulforaphane (4478-93-7/S4441); ethacrynic acid (58-54-8/E1800000); menadione (58-27-5/M9429); oxyphenbutazone (129-20-4/SML0540); papaverine hydrochloride (61-25-6/P3510); phenothiazine (92-84-2/P14831); thimerosal (54-64-8/T5125); thymoquinone (490-91-5/274666); ZPCK (26049-94-5/860794). Compounds were stored at −20 °C and stock solutions for tissue culture and *C. elegans* experiments were freshly prepared at 10–20 mM in DMSO.

**Synthesis of biotinylated papaverine**. Papaverine hydrochloride (1) was mono-demethylated by treatment with sodium ethanethiolate in DMF resulting in a mixture of 6-desmethylpapaverine (2) and pacodine (3). (+)-Biotin *N*-hydroxysuccinimide ester (4) was coupled to 2-(methylamino)ethanol (5) resulting in biotinylated linker control (6). Pacodine (3) and biotinylated linker control (6) were condensed in a ᴍɪᴛꜱᴜɴᴏʙᴜ reaction resulting in biotinylated papaverine (7). Numbers in parentheses correspond to compounds as depicted in Supplementary Fig. 4a. Detailed experimental procedures, references, and analytical data are contained in the Supplementary Methods.

**Protein isolation and immunoblotting**. For *C. elegans* total protein isolation, nematodes collected from three synchronized plates were washed three times with ice-cold S-Buffer and flash-frozen in liquid nitrogen. Frozen pellets were ground in a nitrogen-chilled mortar and suspended in 200 µl Sørensen buffer with 1x Halt Protease and Phosphatase Inhibitor Cocktail (Thermo Fisher Scientific, #78440) and additionally 2 mM sodiumfluoride, 2 mM sodium orthovanadate, 1 mM PMSF, and 2 mM EDTA. Samples were centrifuged for 7 min at 12,000 × *g* and 4 °C to obtain cleared supernatants. Total protein from HEK293 cells was isolated using RIPA buffer with 1x Halt Protease and Phosphatase Inhibitor Cocktail (Thermo Fisher Scientific, #78440). Trypsinized cells were counted, then pelleted by centrifugation for 5 min at 300 × *g* and washed once with ice-cold PBS, before adding 100 µl RIPA buffer per 2 × 10^6 cells. After 5 min incubation on ice, samples were sonicated at 50% amplitude for 3 × 10 s and centrifuged for 15 min at 14,000 × *g* and 4 °C. Supernatants were used for protein quantification with a standard BCA assay and then boiled for 10 min at 95 °C in 1x Laemmli sample buffer. Total protein extracts were used for standard SDS-PAGE and western blotting procedures using the primary and secondary antibodies listed below. The signal was visualized using ECL substrate (Bio-Rad, #1705061). Immunoblot images were acquired with the ChemiDoc MP Imaging System and Image Lab Touch Software V2.4 (Bio-Rad), using the following antibodies: Phospho-p38 MAPK (Thr180/Tyr182) antibody (Cell Signaling, #9211) at 1:1000 dilution with secondary anti-rabbit (Cell Signaling, #7074); α-Tubulin antibody (Sigma-Aldrich, #T6199) at 1:2000 dilution with secondary anti-mouse (Cell Signaling, #7076); FLAG M2 antibody (Sigma-Aldrich, #F3165) at 1:1000 dilution with secondary anti-mouse (Cell Signaling, #7076). All secondary antibodies were used at a dilution of 1:2000.

**RNA extraction and RT-qPCR**. For *C. elegans* total RNA isolation, nematodes collected from three synchronized plates were washed three times with ice-cold S-Buffer and flash-frozen in liquid nitrogen. Afterward, 500 µl ice-cold TRI Reagent (Molecular Research Center, #TR118) was added to the ground pellet. Concentration of the resulting RNA was measured with an LVis plate on a CLARIOstar microplate reader (BMG LABTECH) and stored at −80 °C until further use. Reverse transcription of RNA to cDNA was performed using the High-Capacity cDNA Reverse Transcription Kit together with Oligo(dT) primers following the manufacturer's protocol. Quantitative real-time PCR was carried out on a ViiA 7 Real-Time PCR System using the SYBR Select Master Mix in 384 well plates according to the manufacturer's instructions, and with the recommended cycling conditions. Samples were analyzed by the ddCT method using QuantStudio Real-Time PCR Software V1.3, with normalization to the reference gene Y45F10D.4 for *C. elegans*[64]. Instrument, reagents, and software from Thermo Fisher Scientific. Sequences are listed in Supplementary Table 3.

**Next-generation sequencing (RNA-Seq)**. To identify genes regulated by *C. elegans* GRH-1, RNA-Seq and data analysis was performed using three independent biological samples of total RNA extracted from either wild-type N2 nematodes or from the *grh-1* overexpressing strain used throughout this study. Samples were obtained from synchronized populations, prepared as described below for lifespan assays (at 48 h post-development).

*Library preparation*. The quality of the isolated RNA was determined with a Qubit (1.0) Fluorometer (Life Technologies) and a Bioanalyzer 2100 (Agilent). Only those samples with a 260 nm/280 nm ratio between 1.8–2.1 and a 28S/18S ratio within 1.5–2 were further processed. The TruSeq RNA Sample Prep Kit v2 (Illumina, #RS-122-2001/2) was used in the succeeding steps. Briefly, total RNA samples (100–1000 ng) were poly A enriched and then reverse-transcribed into double-stranded cDNA. The cDNA samples were fragmented, end-repaired, and poly-adenylated before ligation of TruSeq adapters containing the index for multiplexing. Fragments containing TruSeq adapters on both ends were selectively enriched with PCR. The quality and quantity of the enriched libraries were validated using Qubit (1.0) Fluorometer and the Caliper GX LabChip GX (Caliper Life Sciences). Products are a smear with an average fragment size of ~260 bp. The libraries were normalized to 10 nM in Tris-Cl 10 mM, pH 8.5 with 0.1% Tween-20.

*Cluster generation and sequencing*. The TruSeq PE Cluster Kit HS4000 or TruSeq SR Cluster Kit HS4000 (Illumina) were used for cluster generation using 10 pM of pooled normalized libraries on the cBOT. Sequencing was performed on the Illumina HiSeq 2000 single-end at 100 bp using the TruSeq SBS Kit HS4000 (Software: HCS Software for HiSeq 2500, 2000, 1500, and 1000 Systems V2.2.68; Illumina).

*RNA-Seq data analysis*. Bioinformatic analysis was performed using the R package ezRun (github.com/uzh/ezRun) within the data analysis framework SUSHI[65]. Raw reads were quality checked using FastQC and FastQ Screen[66]. Quality controlled reads (adapter trimmed, first and last 4 bases hard trimmed, minimum average quality Q10, minimum tail quality Q10, minimum read length 20 nt) were aligned to the *C. elegans* reference genome (Ensembl WBcel235) using the STAR aligner[67]. Expression counts were computed using featureCounts in the Bioconductor package Subread[68]. Differential expression analysis was performed using the Bioconductor DESeq2 package, where raw read counts were normalized using the quantile method, and differential expression was computed using the quasi-likelihood (QL) F-test[69]. Gene ontology (GO) enrichment analysis was performed using Bioconductor packages GOseq[70] and GOstats[71]. Quality checkpoints, such as quality control of the alignment and count results, were implemented in ezRun and applied throughout the analysis workflow to ensure correct data interpretation.

The datasets generated for this study can be found in the NCBI's Gene Expression Omnibus, GEO Series accession number GSE159077.

*In silico GRH-1 binding sites analysis.* Presence of GRH-1 binding sites in the predicted promoters of differentially expressed genes (DEGs) were determined using the *BiocManager::TFBSTools* function[72] in R and the FIMO algorithm[73] from the MEME Suite[74] in parallel, using a GRH-1 position weight matrix available from literature[15,17]. Both analyses were run on the predicted promoter regions of all DEGs previously determined by RNA-Seq data analysis. Predicted promoter regions were defined as the sequences spanning from 2.5 kb upstream to 1 kb downstream of each DEG's transcriptional start site. Only GRH-1 binding sites identified by both methods were considered further.

**Tissue-specific correlation of human genes with *GRHL1*.** GTEx database (v8) data for tissue-specific gene TPMs (version 2017-06-05_v8_RNASeQCv1.1.9) was obtained from www.gtexportal.org/home/datasets[40]. Pearson's correlations between expression of GRHL1 and all the other genes were calculated with the Python package scipy.stats. Data is visualized with Python packages matplotlib and seaborn.

**GSEA of human genes correlating with *GRHL1*.** In each tissue, Pearson's *r* correlation values of genes with *P*-values <0.05, obtained as described above, were used to calculate gene set enrichment analysis (GSEA) with the Python package gseapy, using gene set files retrieved from download.baderlab.org/EM_Genesets/current_release/Human/symbol (gene set assembly: Human_GO_AllPathways_with_-GO_iea_March_01_2021_symbol.gmt). Data is visualized with Python packages matplotlib and seaborn.

**BXD mice data analysis.** Murine *Grhl1* expression data and phenotypic traits were obtained from GeneNetwork2 (www.genenetwork.org/). Gene expression datasets analyzed: EPFLMouseMuscleCDRMAEx1112, EPFLMouseMuscleCDRMA1211, EPFLMouseMuscleHFDRMAEx1112, EPFLMouseMuscleHFDRMA1211, EPFLMouseLiverCDEx0413, EPFLMouseLiverCDRMA0413, EPFLMouseLiverCDRMA0818, EPFLMouseLiverHFDRMA0818, EPFLMouseLiverHFDRMA0413.

Phenotypic traits analyzed: 21450 (Lifespan); 17661, 17662 (glycemia during oGTT - AUC); 17663, 176634 (insulin during oGTT - AUC); 17603, 17604 (body weight percentage gain 8–28 weeks).

In each expression dataset, *Grhl1* expression RMA in each BXD strain was Z-score normalized, and the average value of *Grhl1* Z-score expression across the same tissue/condition (chow diet muscle, high-fat diet muscle, chow diet liver, high-fat diet liver) in each BXD strains was used to calculate Pearson's *r* correlations with the phenotypic traits of interest. Z-score normalization and Pearson correlation coefficients were computed with Python package scipy.stats. Data is visualized with Python packages matplotlib and seaborn.

**GTEX and FUSION dataset analysis.** Data from Common Fund (CF) Genotype-Tissue Expression Project (GTEx) (phs000424.v8.p2) and The Finland-United States Investigation of NIDDM Genetics (FUSION) Tissue Biopsy Study (phs001048.v2.p1) datasets were obtained from dbGaP (approved request #28650: "Population genetics of human aging").

For the GTEx dataset, only individuals who died of natural causes (field "death manner" DTHMNNR = Natural), were included in the analysis of Spearman rank correlation coefficients for age at death and for *GRHL1* expression in skeletal muscle (ENSG00000134317). Data is represented as the scatterplot of age versus *GRHL1* log2 TPM in skeletal muscle, and the linear regression line is shown to demonstrate correlation.

For the FUSION dataset, Spearman rank correlation coefficients for fasting blood glucose (field GL0) and expression of *GRHL1* (ENSG00000134317) in skeletal muscle were computed in the entire population (All) or in individuals with impaired glucose tolerance (ogtt_status = IGT) or type 2 diabetes (ogtt_status = T2D). Data is represented as the scatterplot of fasting blood glucose versus *GRHL1* log2 FPKM in skeletal muscle, and the linear regression line is shown to demonstrate correlation.

All analyses were performed using the Python package scipy.stats.

**Confocal imaging.** Worms were mounted on freshly prepared agarose pads (3% agarose in S-buffer) and paralyzed using 20 mM tetramisole[75]. Confocal microscopy was performed using an Olympus FluoView 3000 (Olympus Corporation) microscope with inverted stand. Fluoview FV31S-SW software was used for image acquisition. Single plane images of groups of worms were acquired using an UPLFLN 10 × 2 objective (exc/em GFP: 488/510), while Z-stacks of cells were acquired using an UPLSAPO 60X objective (exc/em Alexa 488: 499/520, Mitotracker: 578/598, DAPI: 359/461). Single plane images of cells were overlayed using Fiji ImageJ.

**Immunofluorescence.** Cells were seeded in wells of a 6-well plate, on top of a glass coverslip (VWR, #631-1567) and left to attach overnight. Media was removed and cells were washed once with PBS. Mitotracker Red (ThermoFisher, #M7513) was applied to the cells in DMEM at a final concentration of 250 nM and cells were incubated for 30 min at 37 °C in the incubator. All subsequent steps were performed in the dark. Cells were washed twice with PBS and then fixed with 4% PFA at 37 °C for 15 min. After two washes with PBS, cells were incubated with 0.2%

Triton in PBS for 5 min at RT. Cells were then washed 3 times for 5 min each with PBS on a shaker. Blocking solution (2% BSA and 0.2% Triton in PBS) was added and cells were incubated on a shaker for 30 min at RT. Primary anti-FLAG antibody (Sigma-Aldrich, #F3165) was added at a dilution of 1:200 in blocking solution and cells were incubated for 1 h at 37 °C. Subsequently, cells were washed with PBS three times for 5 min each and secondary Alexa Fluor® 488 goat anti-mouse IgG antibody (ThermoFisher, #A-21141) was added at a dilution of 1:1000 in blocking solution. Afterward, cells were incubated for 1 h at 37 °C. Cells were washed with PBS three times for 5 min each and then coverslips were mounted using DAPI mounting media ProLong Diamond Antifade Mountant with DAPI (Invitrogen, #P36966) and incubated overnight at RT prior to imaging.

***C. elegans* RNAi and compound treatment.** For RNAi-mediated gene knockdown experiments, *E. coli* HT115 bacteria were applied to nematodes by feeding as previously described[76]. The clones for RNAi of *aak-2*/T01C8.1, *daf-16*/R13H8.1, *piki-1*/F39B1.1, *pmk-1*/B0218.3, and *skn-1*/T19E7.2 were obtained from the RNAi ORF library v1.1 (Thermo Fisher Scientific). The clones for *lsd-1*/T08D10.2 and *spr-5*/Y48B1B.6 were obtained from the Ahringer RNAi library (Source BioScience). The clone for *grh-1*/Y48G8AR.1 was constructed as previously described[17]. The clone for *daf-2*/Y55D5A.5 was thankfully provided by the Kenyon lab. Compounds were dissolved in DMSO to a stock concentration of 20 mM and added to NGM agar before pouring. Final DMSO concentration in NGM was 0.5% All compound treatments were performed on HIT OP50. To maintain synchronized populations during long-term experiments, nematodes were washed off the plates into 15 ml tubes every day of the reproductive period, allowed to settle, and then washed again repeatedly until the supernatant was free of progeny. The clean nematodes were then transferred to the freshly prepared treatment plates.

***C. elegans* lifespan assays.** All *C. elegans* lifespan assays, aside from heat shock experiments, were performed at 20 °C according to standard protocols as previously described, explicitly omitting FUdR[77]. Briefly, adult nematodes were allowed to lay eggs for four to 9 h and the resulting eggs incubated for 64 h at 20 °C on NGM agar plates inoculated with OP50 to obtain a synchronized population of young adult nematodes. For a typical lifespan assay, 100 young adult nematodes per condition were manually transferred to NGM agar plates (30–35 nematodes per 55 mm petri dish) inoculated with the respective bacteria as indicated. For the first 10–12 days, nematodes were transferred daily and afterward every 2–3 days. Nematodes showing no reaction to gentle stimulation were scored as dead. Nematodes that crawled off the plates, displayed internal hatching, or a protruding vulva were censored. For heat shock lifespan assays, nematodes on first day of adulthood were transferred to 33 °C for 1 h and then returned to 20 °C. This treatment was repeated every seven days for the remainder of the experiment's duration and wild-type control nematodes were subjected to the same procedure throughout.

***C. elegans* health assays.** To determine fertility, nematodes were synchronized as described above. Single L4 larvae were transferred onto individual plates inoculated with OP50 bacteria (10 plates per condition) and subsequently onto fresh plates every day of the reproductive period. Progeny were allowed to hatch and counted manually.

To determine motility, nematodes were synchronized and single L4 larvae transferred onto individual plates inoculated with OP50 bacteria and subsequently onto fresh plates for two days. Afterward, 15–20 nematodes were transferred to fresh plates and 30 s video clips were recorded (Leica M165FC microscope with Leica camera DFC 3000G). Three independent videos per condition were analyzed regarding the speed of each population. Videos were analyzed using Fiji[78] with the wrMTrck plugin as described[79].

**Luciferase assay compound screening.** Unless stated otherwise, HEK293 luciferase reporter cell lines were seeded into white clear bottom 96-well plates (Greiner Bio-One, #655098) in 100 µl full DMEM and at $5 \times 10^4$ cells/well, 24 h prior to compound treatment. Final DMSO concentration for all treatment and control conditions was adjusted to 0.5%. Readout was performed 24 h after treatment (i.e., in total 48 h after seeding), using a CLARIOstar microplate reader (BMG LABTECH) and the ONE-Glo Luciferase Assay System (Promega Corporation, #E6130) according to the manufacturers' instructions. Details for individual assay setups are as follows:

*Primary compound screening.* Compounds from "The Spectrum Collection" library were pre-diluted from 10 mM stocks and added to wells to a final concentration of 10 µM. Positive control wells were treated with either 5 µM sulforaphane (NRF2/ARE reporter) or 2 µM scriptaid (GRHL reporter), i.e., the same compounds used for initial screening and verification of the reporters. Luciferase activity after compound treatment was normalized to DMSO controls and a two-fold induction of the respective reporter was used as cut-off for primary hit identification.

*Dose–response confirmation of hit compounds.* Resourced individual compounds were used in a three-point dose-response assay at 5, 10, and 20 µM final concentration, with two to three replicate plates at individual timepoints. Overall setup was as for primary compound screening, additionally using the non-lytic CellTiter-Fluor Cell Viability Assay (Promega Corporation, #G6082) according to the

manufacturer's instructions and prior to measuring luciferase activity. Luciferase activity was normalized to cell viability and DMSO controls.

*GRHL1-dependency of hit compounds.* Relative GRHL1-dependency of selected compounds at 5 and 10 μM final concentration was measured with a wild-type GRHL reporter (GRHL[WT-LUC]) and a GRHL reporter with a CRISPR/Cas9-mediated *GRHL1* deletion (GRHL[NULL-LUC]), using two individual replicates per cell line and concentration. Luciferase activity was normalized to the respective DMSO control and relative GRHL1-dependency determined as GRHL[NULL-LUC]/GRHL[WT-LUC] fold change across both concentrations and two independent GRHL[NULL-LUC] cell lines.

*Impact of GRHL1 mutation on GRHL reporter activation.* GRHL reporter cells were seeded at $1 \times 10^4$ cells/well and otherwise as above. After 24 h, cells were individually transfected with either a wild-type or mutated *GRHL1* overexpression vector or an empty vector control, using the Lipofectamine 3000 Transfection Reagent (Thermo Fisher Scientific, #L3000008). Readout was performed 48 h after transfection (in total 72 h after seeding). Two independent experiments with in total 4 individual replicate wells per condition were performed and activity determined as percentage change compared to transfection with the vector encoding wild-type GRHL1, normalized to the respective empty vector control sample.

### Tandem-affinity purification of human GRHL1 from nuclei
*Nuclei isolation.* Wild-type HEK293 cells or a HEK293 cell line overexpressing C-terminally 3xFLAG-6xHis tagged wild-type human GRHL1 (GRHL1[WT-TAG]) were seeded into 15 cm plates at a split ratio of 1:8 and treated with 20 μM papaverine or 0.1% DMSO as vehicle control 24 h prior to the isolation procedure. For nuclei isolation, cells were trypsinized, counted, and $20 \times 10^6$ cells per sample transferred to a 50 ml tube. Cells were pelleted by centrifugation for 3 min at $1000 \times g$, washed once with ice-cold PBS, and pelleted again. Cell pellets were resuspended in 1 ml freshly prepared, filter-sterilized ice-cold hypotonic lysis buffer (0.01 M Tris-HCl pH 7.4, 0.01 M NaCl, 10% Glycerol, 0.3% NP-40, 1x Halt Protease and Phosphatase Inhibitor Cocktail from Thermo Fisher Scientific, #78440) and incubated on ice for 15 min. Afterward, nuclei were pelleted by centrifugation for 8 min at $800 \times g$ and 4 °C, the supernatant containing the cytoplasmic fraction discarded, and nuclear pellets flash-frozen in liquid nitrogen and immediately stored at −80 °C until tandem-affinity purification.

*3xFLAG and 6xHis tandem-affinity purification.* Nuclei isolated from GRHL1[WT-TAG] overexpressing cells and wild-type HEK293 cells were subjected to the same purification procedures, with the latter samples serving as background control for unspecific absorption of proteins to the utilized affinity matrices. For 3xFLAG purification, nuclear pellets were resuspended in 1 ml freshly prepared, filter-sterilized ice-cold FLAG lysis buffer (0.05 M Tris-HCl pH 7.4, 0.15 M NaCl, 10% Glycerol, 1% Triton X-100, 0.01% Tween-20, 1x Halt Protease and Phosphatase Inhibitor Cocktail from Thermo Fisher Scientific, #78440), briefly vortexed, and incubated on ice for 15 min. Afterward, samples were centrifuged for 10 min at $14,000 \times g$ and 4 °C, and supernatants used as input for purification. Anti-FLAG M2 Magnetic Beads (Millipore/Merck KGaA, #M8823) were prepared in TBS according to the manufacturer's instructions and 30 μl packed gel volume used per reaction. Supernatants were added to beads and incubated for 2 h at 4 °C with end-over-end rotation. After incubation, supernatants were discarded, and beads washed twice with 300 μl TBS + 10% Glycerol. Proteins were eluted with 150 μl His binding buffer (0.02 M HEPES-KOH pH 7.4, 0.3 M NaCl, 10% Glycerol, 0.02 M Imidazol-HCl pH 7.4, 0.01% Tween-20, 1x Halt Protease and Phosphatase Inhibitor Cocktail from Thermo Fisher Scientific, #78440) containing 150 ng/μl 3X FLAG peptide (Millipore/Merck KGaA, #F4799) by incubation for 30 min at 4 °C with end-over-end rotation. Supernatants with eluted proteins were adjusted to 700 μl with His binding buffer and added to 50 μl Dynabeads His-Tag Isolation and Pulldown (Thermo Fisher Scientific, #10104D) per reaction. After another incubation for 30 min at 4 °C with end-over-end rotation, Dynabeads were washed twice with 300 μl His binding buffer and once with 300 μl His binding buffer without Tween-20. During the last washing step, samples were transferred to new reaction tubes, any remaining supernatants discarded completely, and beads resuspended in 100 μl His binding buffer without Tween-20 per sample.

*On-bead digestion.* For each sample, the beads were washed once with 100 μl of digestion buffer (10 mM Tris + 2 mM CaCl₂, pH 8.2). The washed beads were resuspended in 90 μl digestion buffer, and the proteins were on-bead digested using 10 μl of Sequencing Grade Trypsin (100 ng/μl in 10 mM HCl, Promega, #V5111). The digestion was carried out in a microwave instrument (Discover System, CEM) for 30 min at 5 W and 60 °C. The supernatants were collected, and the peptides were extracted from the beads with 100 μl of TFA-buffer (0.1% TFA in 10 mM Tris, 2 mM CaCl₂, 50% acetonitrile). The supernatants were combined, and the samples were finally dried in a speed-vac, resolubilized in 20 μl of 0.1% formic acid and centrifuged at max speed ($20,000 \times g$) for 10 min. Ten microliters of digested peptides were injected for shotgun liquid chromatography with tandem mass spectrometry (LC-MS/MS) as described below.

### Protein pull-down using biotinylated papaverine
*Cell lysis.* Cells were collected from 15-cm dishes without trypsinization. Cells were centrifuged at $500 \times g$, 4 °C for 5 min and washed once with cold PBS. Cells were

spun down again as above, and supernatant was discarded. Lysis buffer (20 mM HEPES pH 7.3, 50 mM KCl, 5 mM MgCl₂, 0.01% NP40, 2 mM NaF, 2 mM Na₃VO₄) was added, cells were vortexed and incubated on ice for 15 min. Cells were sonicated 3 times for 2 s each at 50% amplitude and incubated on ice for 10 min. The lysate was centrifuged at $13,000 \times g$ for 5 min and 4 °C. Supernatant was collected and protein concentration was measured using a standard BCA assay.

*Bead preparation and pull-down.* For each condition, 40 μl of streptavidin bead-suspension was used (Thermo Fisher Scientific, #11205D). Beads were equilibrated three times with lysis buffer and subsequently incubated with biotinylated papaverine (compound 7 in Supplementary Fig. 4a) or biotinylated linker control (compound 6 in Supplementary Fig. 4a) for 30 min at RT with rotation. Cell lysate was first incubated with biotinylated linker control for 3 h at 4 °C with rotation, then split into half and incubated with either biotinylated-linker-beads (as a control condition) or biotinylated-papaverine-beads for 4 h at 4 °C with rotation. After incubation, beads were washed five times with lysis buffer (w/o NP40, NaF, or Na₃VO₄) and stored at −80 °C prior to MS analysis. All experiments were performed in triplicate.

*On-bead digestion.* A variation of the on-bead digestion protocol from the Mann lab[80] was used. Proteins were pre-digested at 30 °C for 15 min by adding 20 μl elution buffer (2 M urea in 50 mM Tris-HCl, pH 8.0) supplemented with 5 mM DTT and 100 ng Sequencing Grade Trypsin (Promega, #V5111). Next, iodoacetamide was added to a final concentration of 3 mM, and incubation was continued for another 15 min. Subsequently, proteins were eluted by adding two times (50 and 30 μl) elution buffer supplemented with 1 mM DTT and 5 ng/μL trypsin. Eluates were transferred to new microcentrifuge tubes, incubated at 32 °C overnight (or minimum 6 h). The digestion was stopped by adding 0.5% of TFA. All incubation steps were performed in a thermoshaker at 400 rpm gentle shaking. Digested peptides were desalted using self-packed C18 Stage-Tips[81], vacuum dried, and resolubilized in 20 μl 3% acetonitrile, 0.1% formic acid for mass spectrometry analysis. Four microliters of digested peptides were injected for shotgun liquid chromatography with tandem mass spectrometry (LC-MS/MS) as described below.

### Liquid chromatography-mass spectrometry analysis
Label-free shotgun LC-MS/MS was performed either on a Q Exactive mass spectrometer (Software: Exactive MS Series Instrument Control Software V2.9; Thermo Fisher Scientific) equipped with a nanoAcquity UPLC system (Waters Inc.) for the purified GRHL1[WT-TAG] and associated control samples, or on an Orbitrap Fusion Lumos mass spectrometer (Software: Orbitrap Tribrid MS Series Instrument Control Software V3.1; Thermo Fisher Scientific) equipped with an Acquity UPLC M-Class system (Waters Inc.) for the purified biotinylated papaverine and associated control samples. Peptide mixtures underwent reverse-phase chromatographic separation before nano-electrospray ionization (ESI) with a Digital PicoView nanospray source (New Objective). Peptides were eluted at a flow rate of 300 nl/min with a binary acetonitrile gradient system composed of (A) 0.1% formic acid in H₂O and (B) 0.1% formic acid in acetonitrile. Gradient was set with the following conditions:

Q Exactive: BEH300 C18 column (1.7 μm, 75 μm × 150 mm, Waters Inc.); multistep gradient of 5–35% B in 90 min, 40% B in 5 min, and 80% B in 1 min.

Orbitrap Fusion Lumos: HSS C18 T3 Col 100A column (1.8 μm, 75 μm × 250 mm, Waters Inc.); multistep gradient of 5–35% B in 135 min, 40% B in 5 min, and 80% B in 1 min.

For the data-dependent analysis, mass spectrometers were set as follows:

Q Exactive: precursor scan range of 350–1500 *m/z* at a resolution of 70,000 at 200 *m/z* with AGC (automated gain control) target value of $3 \times 10^6$, followed by HCD (higher-energy collision dissociation) fragmentation on the 12 most intense signals per cycle with a normalized collision energy of 25.

Orbitrap Fusion Lumos: precursor scan range of 300–2000 *m/z* at a resolution of 120,000 at 200 *m/z* with AGC target value $4 \times 10^6$; followed by HCD fragmentation on the 15 most intense signals per cycle, with a collision energy of 35.

In both cases, only precursors with intensity above 25,000 were selected for MS/MS. Precursor masses previously selected for MS/MS measurement were excluded from further selection for 30 s (Orbitrap Fusion Lumos) or 40 s (Q Exactive), and the exclusion window was set at 10 ppm. All measurements were acquired using internal lock mass calibration on *m/z* 371.1010 and 445.1200.

*Data analysis of GRHL1[WT-TAG] samples.* For the identification of interactors, the raw data were converted into Mascot Generic Format files (.mgf) using ProteoWizard (http://proteowizard.sourceforge.net/), and the proteins were identified using the Mascot search engine (Matrix Science, version 2.5.1.3). Spectra were searched against a UniProtKB/Swiss-Prot *Homo sapiens* proteome database (taxonomy 9606, release 2018_09 [https://ftp.uniprot.org/pub/databases/uniprot/previous_major_releases/release-2018_09/knowledgebase]), concatenated to its reversed decoyed FASTA database. Methionine oxidation was set as variable modification, and enzyme specificity was set to trypsin allowing a maximum of two missed cleavages. A fragment ion mass tolerance of 0.030 Da and a parent ion tolerance of 10.0 PPM were set. Scaffold (Proteome Software Inc., version 4.10) was used to validate MS/MS-based peptide and protein identifications. Peptide identifications were accepted if they achieved a false discovery rate (FDR) of <0.1% by

the Scaffold Local FDR algorithm. Protein identifications were accepted if they achieved an FDR of <1.0% and contained at least 2 identified peptides.

The search engine PEAKS 8.5 (PEAKS X, Bioinformatic Solutions) was used for the characterization of post-translational modifications (PTMs) on protein GRHL1$^{WT-TAG}$. Data were searched with a fragment ion mass tolerance of 0.03 Da and a parent ion tolerance of 15.0 PPM. Oxidation (M), Acetylation (K), Dimethylation (KR), Methylation (KR), Phosphorylation (STY), and Ubiquitination (K) were specified as a variable modification. A maximum of 3 missed cleavages were allowed. Peptide identifications were accepted if they achieved a peptide false discovery rate (FDR) of <0.1% and the proteins contained at least 2 identified peptides.

Detected GRHL1$^{WT-TAG}$ PTMs were considered as papaverine-specific if the MS/MS spectra could be identified in at least one of the papaverine-treated samples and not in any of the DMSO control samples. Proteins were considered as GRHL1 interactors if present in at least 2/3 samples GRHL1$^{WT-TAG}$ and either fully absent or enriched at least ≥3-fold over the respective control, based on sample average of unique peptide counts. In total 130 proteins were identified as potential GRHL1$^{WT-TAG}$ interaction partners. These proteins were curated manually, and each assigned a non-exclusive category and classification reflecting its most prominent cellular function, based on associated gene ontology terms and information available from UniProtKB/Swiss-Prot[82] (Supplementary Data 5).

The mass spectrometry proteomics data have been deposited to the ProteomeXchange Consortium via the PRIDE[83] partner repository with the dataset identifier PXD021768.

*Data analysis of biotinylated papaverine pull-down samples.* All raw data analyses were handled and annotated by the local laboratory information management system (LIMS) (https://doi.org/10.1145/1739041.1739135), further analyzed with MaxQuant software suite version 1.6.3.3 (Max Planck Institute of Biochemistry, Munich) supported by the Andromeda search engine[84]. Spectra were searched against the UniProt Human proteome database encompassing 75,004 protein entries (proteome ID UP000005640, version from 2019-05-05), concatenated to its reversed decoyed FASTA database. Data were searched with carbamidomethylation as a fixed modification and protein N-terminal acetylation and methionine oxidation as variable modifications. A maximum of two missed cleavages were allowed while requiring strict trypsin specificity, and only peptides with a minimum sequence length of six were considered for further data analysis. Only peptides and proteins with a false discovery rate of <1% were accepted.

Proteins were considered as papaverine interactors if present in at least 2/3 replicates of the biotinylated papaverine samples and enriched ≥3-fold over control, based on sample averages of both label-free quantification intensities and unique peptide counts.

The mass spectrometry proteomics data have been deposited to the ProteomeXchange Consortium via the PRIDE[83] partner repository with the dataset identifier PXD021808.

**Statistical analyses.** Statistical analyses for all data except those from lifespan assays were carried out using a *t*-test with appropriate parameters, i.e., a two-tailed unequal variances *t*-test used for comparison of the unpaired control vs. treatment groups. For comparing distributions between different groups in the lifespan assays, statistical calculations were performed using JMP software V9.0 (SAS Institute), applying the log-rank test. All other calculations were performed using Microsoft Excel 365 or GraphPad Prism V8.2.0 (GraphPad Software). *P*-values are reported in detail, without the use of arbitrary star ratings. However, in line with longstanding conventions used in the field, *P*-values <0.05 are generally referred to as indicating a significant difference, i.e., a rejection of the null hypothesis.

**Reporting summary.** Further information on research design is available in the Nature Research Reporting Summary linked to this article.

## Data availability

Publicly accessible databases and datasets used in this study are as follows and in the order they are referenced in the Methods section. RNA-Seq data analysis: *C. elegans* reference genome (Genome assembly WBcel235). Tissue-specific correlations of human genes with *GRHL1*: GTEx database (v8) data for tissue-specific gene TPMs (version 2017-06-05_v8_RNASeQCv1.1.9). GSEA of human genes correlating with *GRHL1*: Gene set assembly Human_GO_AllPathways_with_GO_iea_March_01_2021_symbol.gmt retrieved from [download.baderlab.org/EM_Genesets/current_release/Human/symbol]. BXD mice data analysis: Murine *Grhl1* expression data and phenotypic traits were obtained from GeneNetwork2 [www.genenetwork.org]. Accession codes of gene expression datasets analyzed (accessed on 06.03.2021—detailed retrieval procedure can be found here: https://github.com/araldi/Grigolon-et-al_GRHL1_NatureCommunications/blob/main/BXD/1_Download_BXD_datasets.ipynb): EPFLMouseMuscleCDRMAEx1112 and EPFLMouseMuscleCDRMA1211 (muscle chow diet), EPFLMouseMuscleHFDRMAEx1112, EPFLMouseMuscleHFDRMA1211 (muscle high fat diet), EPFLMouseLiverCDEx0413, EPFLMouseLiverCDRMA0413, EPFLMouseLiverCDRMA0818 (liver chow diet); EPFLMouseLiverHFDRMA0818, EPFLMouseLiverHFDRMA0413, EPFLMouseLiverHFCEx0413 (liver high fat diet). Phenotypic traits analyzed (accessed on

06.03.2021—detailed retrieval procedure can be found here: https://github.com/araldi/Grigolon-et-al_GRHL1_NatureCommunications/blob/main/BXD/1_Download_BXD_datasets.ipynb): 21450 (Lifespan); 17661, 17662 (glycemia during oGTT - AUC); 17663, 176634 (insulin during oGTT - AUC); 17603, 17604 (body weight percentage gain 8–28 weeks). GTEX and FUSION dataset analysis: Common Fund (CF) Genotype-Tissue Expression Project (GTEx) (phs000424.v8.p2) and The Finland-United States Investigation of NIDDM Genetics (FUSION) Tissue Biopsy Study (phs001048.v2.p1) datasets were obtained from dbGaP (request #28650: "Population genetics of human aging"). Mass spectrometry data analysis: UniProtKB/Swiss-Prot *Homo sapiens* proteome database (taxonomy 9606, release 2018_09 [https://ftp.uniprot.org/pub/databases/uniprot/previous_major_releases/release-2018_09/knowledgebase/]). The RNA-Seq data generated in this study have been deposited in NCBI's Gene Expression Omnibus (GEO) database under GEO Series accession number GSE159077. The mass spectrometry proteomics data generated in this study have been deposited to the ProteomeXchange (PX) Consortium via the PRIDE database under dataset identifiers PXD021768 and PXD021808. All other data supporting the findings of this study are available within this paper, its Supplementary Information, and its Supplementary Files. Source data are provided with this paper.

## Code availability

The codes used for the analyses of BXD, FUSION, and GTEX data are available at https://github.com/araldi/Grigolon-et-al_GRHL1_NatureCommunications.

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

## Acknowledgements

*C. elegans* strains used in this work were provided by the Caenorhabditis Genetics Centre (Univ. of Minnesota, USA), which is funded by NIH Office of Research Infrastructure Programs (P40 OD010440). This study was funded by the Swiss National Science Foundation (#31003A_176127 to M.R.). The Ristow laboratory is further supported by the Horizon 2020 program of the European Union (Ageing with Elegans, #633589). The authors thank Cynthia Kenyon for providing the *daf-2* RNAi clone, as well as Sarah Tailly and Uli Kodjadjiku for help with supporting experiments. The authors thank Laura Kunz and Paolo Nanni of the Functional Genomics Center Zurich (FGCZ) for their support in acquisition, analysis, and presentation of mass spectrometry data.

## Author contributions

M.R. and F.F. conceived and supervised the project. G.G. and F.F. performed the majority of experiments, and J.Y.W., C.T., B.L., D.P., and K.Z. performed additional experiments. R.E. and E.C. were responsible for synthesis of biotinylated papaverine. E.A. performed all BXD, GTEx, and FUSION dataset analyses. G.G. and F.F. analyzed the majority of all other data and M.L.F., J.Y.W., and R.E. performed additional data analyses. F.F. and G.G. prepared all figures, with contributions from E.A. and R.E., and M.R. provided additional input. F.F., G.G., E.A., M.S., and M.R. co-wrote the manuscript with input from all authors.

## Competing interests

The authors declare no competing interests.

## Additional information

**Peer review information** *Nature Communications* thanks Ole Jensen and the other anonymous reviewer(s) for their contribution to the peer review this work. Peer reviewer reports are available.

