## [Peer Review File · Nature Communications]

Reviewers' Comments:

Reviewer #1:

Remarks to the Author:

In this manuscript entitled "An evolutionarily conserved transcriptional regulator linked to metabolic control of healthspan", Ristow and colleagues found that papaverine promotes longevity via activating the transcription factor Grainyhead 1 (GRH-1/Grhl1/GRHL1) in multiple species, ranging from *C. elegans* to mammals. Following up their previous paper (Rožanov, L. et al. 2020), the authors showed that a mild increase in *grh-1* using transgenesis increased lifespan, motility, and pathogen resistance in *C. elegans*. By performing a drug screen using a reporter system in cultured human cells, the authors found that papaverine activated GRHL1, and subsequently that papaverine promoted longevity in *C. elegans*. By using proteomic analysis, they showed that post-translational methylation at a lysine residue GRHL1 was required for its activation. They also identified proteins that bound papaverine, including phosphoinositide 3-kinase (PI3K) PIK3C2A. Genetic inhibition of PIK3C2A/piki-1 increased GRHL1 activity in cultured human cells and increased lifespan in *C. elegans*. They further showed that *grh-1* was upregulated in long-lived mutant *C. elegans* with reduced insulin/IGF-1 signaling, in which PI3K acts, while *grh-1* RNAi decreased the longevity. The authors then determined the expression levels of Grhl1 in BXD mice and showed that the Grhl1 expression levels positively correlated with mean lifespan. In addition, GRHL1 expression levels positively correlated with those of insulin signaling-associated genes in human skeletal muscle and the liver. Similar to mice, GRHL1 expression levels also negatively correlated with aging and fasting blood glucose levels in human GTEx cohort. Overall, this paper reports very interesting findings regarding Grainyhead1, an evolutionarily conserved aging-regulating transcription factor, and its potential as a therapeutic target for aging using *C. elegans* and mammalian cells. Following are my concerns that the authors need to address.

Major comments

- 1) The title of this paper is imprecise and too inclusive because "metabolic control of healthspan" seems to be not the main content. The authors did not measure healthspan, the healthy period during life. In addition, the authors did not examine metabolic changes caused by genetic or pharmacological manipulation of *grh-1*. Therefore, the title is misleading and I suggest that the authors revise the title to describe their findings more precisely. Similarly, the definition of healthy lifespan is relatively ambiguous in Introduction.
- 2) One of the key data in this study is a proper and mild increase (about 2 fold) in the level of *grh-1* promoted longevity in *C. elegans*, but strong overexpression did not. They used one hsp-16.2 promoter-driven *grh-1* transgenic line with or without heat shock to claim that. However, I don't think it is sufficient to support their conclusion because it is just one transgenic line and heat shock may have affected the health of the transgenic animals in a different way. I therefore suggest that the authors need to generate additional transgenic lines using different promoters, including its own *grh-1* promoter. By analyzing the expression levels of *grh-1* and the lifespan using multiple lines of transgenic animals, the authors will be able to conclude better for their claim.
- 3) Discussion is mostly reiteration of the Results and I think it should be rewritten to convey the significance and limitation of this study and perspectives.

Minor comments

- 1) In Introduction, it will be better to explain the importance of identifying drug targets regulating aging across species.
- 2) The author should explain why the human GRHL reporter contains the ARE and cite more articles about NRF2. On page 6, please specify the 15 compounds with the highest activation of GRHLWT-LUC "among 210 compounds".
- 3) Please elaborate experimental conditions regarding generation of transgenic *C. elegans* (e.g. what concentration of pDEST-MB14 vector was used for microparticle bombardment?) or heat-shock conditions.
- 4) Please explicitly mention the sample size bigger than three, although the authors present "sample sizes as indicated by depicted individual data points."
- 5) In figure 1d, the authors need to show *C. elegans* fluorescence image into separate panels and magnify images to show nuclear localization of GRH-1 more clearly.
- 6) Figure 3a appears to be more suitable in figure 2 than in figure 3. Please move figure 3a to figure 2.

- 7) In Fig. 4b, please visualize Venn diagram to correctly describe the experimental scheme as GRHL1WT-TAG interacting proteins detected in both DMSO- and papaverine-treated conditions.
- 8) For Fig. 4c, it will be better to show whether the 7 PTM sites are conserved in *C. elegans* as well. In particular K116. If not, they need to discuss what that means in the discussion.
- 9) In Fig. 4e, instead of darker edge colors, using different thickness of edges will be more intuitive.
- 10) Please carefully distinguish *C. elegans* *daf-2* mutations, which confer many beneficial effects, from impaired glucose tolerance in humans.
- 11) Please differently mark individual points in Supplementary Fig. 2a as in Supplementary Fig. 2d. In Supplementary Fig. 2c legend, use "gene set enrichment analysis", not "gene sets enrichment analysis."
- 12) It will be better to perform GSEA of upregulated genes in *grh-1* OEx under conditions related to human GRHL1 among different tissues to better compare the relationship between *C. elegans* GRH-1 and human GRHL1.
- 14) In Supplementary Fig. 3 legend, use (a), not (A).
- 15) In Supplementary Table 3, please mark which genes are related to MAP kinase activity and annotate gene name in the "FIMO - MEME Suite" table.
- 16) The authors mentioned that "The strongest activation of the GRHL reporter across the range of concentrations tested was observed with the compounds 2',4'-Dihydroxy-4-methoxychalcone (DHM-142 chalcone, 286.2 ± 36.6 at $20 \mu\text{M}$, $P = 0.008$) and alantolactone (131.5 ± 5.6 at $20 \mu\text{M}$, $P = 0.017$)" for describing Fig. 2b. However, it seems that vorinostat elicited the greatest effects across the range of concentrations. The authors should elaborate this sentence to better describe the data.

Reviewer #2:

Remarks to the Author:

Grigolon et al describe a study of the *C. elegans* transcription factor GRH1 (human GRHL1). They report the effects of a panel drugs upon GRH1 activation, identify interacting proteins and some PTMs, and provide evidence for links to protein regulation by methylation/demethylation and modulation by insulin/IGF-1 signaling network.

Overall, this is a very well described study and results that provides novel information on aging and metabolism.

The differential GRH1 protein interaction results leads to the conclusion that protein methyl transferases (KMTs) and demethylases (KMDs) might be implicated in the regulation of GRH1. It is common practice to pursue reverse-interactome experiments to confirm the key interacting proteins. However, this was not attempted here. The authors should explain why this obvious experiment was omitted. (The authors do mention that KMT2C and KMT2D were found to interact with GRHL2, but this is not a proof in the context of the present study).

One key aspect of the study is the putative regulation of the GRH1 by PTMs, particularly methylation. I acknowledge that the proteomics data is available via PRIDE, however, it is difficult to retrieve specific spectra of distinct peptides. I want to see the tandem mass spectra of the modified peptides as part of the supplementary materials. Please include figures of the annotated MS/MS spectra for all the modified peptides, including R9me2 and K116me2. Also, include the LC-MS traces (TIC) to demonstrate the quantitative differences between the unmodified and modified peptides, +/- papaverine.

This data is relevant for establishing the role of methylation in regulation of GRH1.

The affinity enrichment using biotin-papaverine is an elegant experiment (suppl. Fig 4). Is the binding mode of papaverine to proteins known? I wonder whether the position of biotin relative to papaverine will play a role, i.e. are there steric hindrances that might affect the way that proteins recognize/bind to papaverine? A short comment on this in the paper will suffice.

Reviewer #3:

Remarks to the Author:

In the provided manuscript, Grigolon et al. follow up on their previous finding that the activity of the transcription factor grainyhead 1 is a transcription factor controlling longevity. They show that overexpression of GRHL1 is sufficient to extend lifespan (assuming a mistake in Fig1h). They then conduct a cell culture-based chemical screen using a repurposing library. They identify inducers of GRHL1 expression and identify vorinostat, papaverine, and a series of other compounds.

Consistent with their hypothesis that *grh1* is a regulator of longevity, some of the identified hits indeed extend lifespan and, most importantly, depend on GRHL1. They then show that these longevity compounds change post-translational modifications on GRHL1. Furthermore, the transcriptional activity of GRHL1 depends on methylation and methyltransferases. Finally, they correlate their findings to mouse and human gene expression datasets suggesting that GRHL1 may indeed be a valuable drug target to treat age-related diseases.

The manuscript is a good example of cross-species drug discovery and how chemicals can effectively work across multiple experimental models to identify evolutionarily conserved mechanisms modulating aging. The authors use a wide variety of different technologies to investigate the underlying mechanisms. The work is thoroughly done. I believe that this manuscript will be of general interest, both to researchers working in the field of aging and to researchers working in drug discovery. The manuscript is clearly written, well thought out, and easy to understand.

There are, however, a few minor issues that have to be addressed or answered. Once these are addressed, it will make an interesting and exciting publication.

Major:

- I don't think there are any major issues.

Minor:

- Line 55: please indicate the *C.elegans* designation for KDM1A and KMTC2 to make it easier for both mammalian and worm geneticists to read and connect to their field of study.

- Fig. 4h. did the authors try each demethylates individually? Are both at the same time necessary? Please clarify

- The lifespan curves for the +HS in Figure 1e and + HS and supplementary Figure 1e look extremely similar. Please confirm that this is not an error. Since they are so similar it could be that the same lifespan curve was accidentally used for both figures but labeled to represent two different overexpression lines.

- Table S2 only reveals upregulated genes in the *grh-1* OE strain. The manuscript never mentions where the entire data set is available or how it will be made available?

- Line 102 states that *pmk-1* is the highest expressed gene from the MAP kinase pathway. However, table S2 shows that *pmk-3* is expressed higher (1.078). So, is *pmk-3* not in the MAP kinase pathway?

- Table S3 contains binding sites for 102 genes for *grh-1*. The gene list S3 completely overlaps with the 242 upregulated genes in S2. That suggests that S3 was constructed by only searching for binding sites in upregulated genes. If that is correct, it should be stated to enable the reader to interpret the list.

- Table S5 (interactome) and Table S7 (papaverine pull-down) share a CPVL as a common component linking papaverine directly to GRHL1. It seems to me that this may be an exciting find. Any specific reason on why this was not followed up?

REVIEWER COMMENTS

Reviewer #1 (Remarks to the Author):

In this manuscript entitled “An evolutionarily conserved transcriptional regulator linked to metabolic control of healthspan”, Ristow and colleagues found that papaverine promotes longevity via activating the transcription factor Grainyhead 1 (GRH-1/Grhl1/GRHL1) in multiple species, ranging from *C. elegans* to mammals. Following up their previous paper (Rozanov, L. et al. 2020), the authors showed that a mild increase in *grh-1* using transgenesis increased lifespan, motility, and pathogen resistance in *C. elegans*. By performing a drug screen using a reporter system in cultured human cells, the authors found that papaverine activated GRHL1, and subsequently that papaverine promoted longevity in *C. elegans*. By using proteomic analysis, they showed that post-translational methylation at a lysine residue GRHL1 was required for its activation. They also identified proteins that bound papaverine, including phosphoinositide 3-kinase (PI3K) PIK3C2A. Genetic inhibition of PIK3C2A/*piki-1* increased GRHL1 activity in cultured human cells and increased lifespan in *C. elegans*. They further showed that *grh-1* was upregulated in long-lived mutant *C. elegans* with reduced insulin/IGF-1 signaling, in which PI3K acts, while *grh-1* RNAi decreased the longevity. The authors then determined the expression levels of Grhl1 in BXD mice and showed that the Grhl1 expression levels positively correlated with mean lifespan. In addition, GRHL1 expression levels positively correlated with those of insulin signaling-associated genes in human skeletal muscle and the liver. Similar to mice, GRHL1 expression levels also negatively correlated with aging and fasting blood glucose levels in human GTEx cohort. Overall, this paper reports very interesting findings regarding Grainyhead1, an evolutionarily conserved aging-regulating transcription factor, and its potential as a therapeutic target for aging using *C. elegans* and mammalian cells. Following are my concerns that the authors need to address.

The authors appreciate the enthusiasm of the reviewer regarding the initially submitted version of the manuscript and its scientific relevance.

Major comments

1) The title of this paper is imprecise and too inclusive because “metabolic control of healthspan” seems to be not the main content. The authors did not measure healthspan, the healthy period during life. In addition, the authors did not examine metabolic changes caused by genetic or pharmacological manipulation of *grh-1*. Therefore, the title is misleading and I suggest that the authors revise the title to describe their findings more precisely. Similarly, the definition of healthy lifespan is relatively ambiguous in Introduction.

Authors’ Response:

The Authors thank Reviewer #1 for this relevant comment. We believe the now implemented changes should help to better describe our key findings both in the Title and Introduction.

We have now changed the title, from “*An evolutionarily conserved transcriptional regulator linked to metabolic control of healthspan*” to “*Grainyhead 1 acts as a drug-inducible conserved transcriptional regulator linked to insulin signaling and lifespan*”, which we find more accurately reflects the most important findings of our paper. We do hope Reviewer #1 agrees but are of course open to additional suggestions.

In addition, we have introduced corresponding minor changes in phrasing and additional clarifications in the Abstract (**lines 26, 34**) and in the Introduction (**lines 57-58 and 72**).

2) One the key data in this study is a proper and mild increase (about 2 fold) in the level of *grh-1* promoted longevity in *C. elegans*, but strong overexpression did not. They used one *hsp-16.2* promoter-driven *grh-1* transgenic line with or without heat shock to claim that. However, I don't think it is sufficient to support their conclusion because it is just one transgenic line and heat shock may have affected the health of the transgenic animals in a different way. I therefore suggest that the authors need to generate additional transgenic lines using different promoters, including its own *grh-1* promoter. By analyzing the expression levels of *grh-1* and the lifespan using multiple lines of transgenic animals, the authors will be able to conclude better for their claim.

Authors' Response:

The Authors thank Reviewer #1 for this comment. The now implemented changes hopefully help to clarify this key claim in our manuscript.

Specifically, we had already analyzed *two independently obtained and independently backcrossed* nematodal lines now named "*grh-1* OEx line 1" and (as before) "*grh-1* OEx line 2", specifically as it pertains to the initial key lifespan experiments that used these two lines in parallel. Both carry identical constructs, namely the *hsp16.2* promoter followed by the *grh-1* cDNA in-frame with a sequence encoding GFP.

One of these two lines (*grh-1* OEx line 1) is (and was) depicted in the main *Figure 1*, where *Panels c to e* depict experiments in the absence of a heat shock, and *Panels f to h* depict the clonal strain in the presence of a heat shock, as indicated.

The second of these two lines (*grh-1* OEx line 2) is (and was) depicted in the *Supplementary Figure 1*, where now *Panels b and c* depict experiments in absence of a heat shock, and now *Panels e and f* depict the clonal strain in the presence of a heat shock, as indicated.

Within the text of the Results paragraph, we now have aimed to further clarify regarding this point and the use of two independent *grh-1* OEx lines for the initial RT-qPCR and lifespan experiments. (paragraph of line 85 onward).

Additionally, we have now also added newly obtained data for a *grh-1* overexpressing strain in which expression is driven by its endogenous *grh-1* promoter (newly added Supplementary Fig. 1h, i). As now also stated in the corresponding Results paragraph, this strain further supports dose-dependency of the effect of *grh-1* overexpression, i.e., only limited overexpression being beneficial while intermediate or strong overexpression is not (lines 102-105).

Regarding the comment of Reviewer #1 that "heat shock may have affected the health of the transgenic animals in a different way": We have now clarified that for lifespan assays with heat shocked *hsp-16.2p::grh-1::gfp* animals (lines 1 and 2), wild-type control animals were subjected to the same treatment (i.e., heat shock), thereby excluding any unspecific effects of the heat shock itself (lines 95-97).

3) Discussion is mostly reiteration of the Results and I think it should be rewritten to convey the significance and limitation of this study and perspectives.

Authors' Response:

We thank Reviewer #1 for this comment and have now tried to limit restatement of Results in the Discussion section and expanded several points, also taking into account the comments of the other Reviewers, to hopefully better convey the significance and limitations of our study (all altered

sections marked with track changes and color highlighting). However, we find the claim that the original Discussion was “**mostly reiteration of the Results**” to not be quite justified.

Minor comments

1) In Introduction, it will be better to explain the importance of identifying drug targets regulating aging across species.

Authors’ Response:

We find this to be an excellent suggestion to improve clarity and have accordingly further explained this point in the Introduction (**lines 41-46**).

2) The author should explain why the human GRHL reporter contains the ARE and cite more articles about NRF2. On page 6, please specify the 15 compounds with the highest activation of GRHLWT-LUC “among 210 compounds”.

Authors’ Response:

The human HEK293 GRHL reporter, and the human HEK293 antioxidant response element (ARE)/NRF2 reporter are two entirely independent cell lines. I.e., the HEK293 human GRHL reporter does not contain the ARE. We have clarified this in the revised version of the results section (**lines 149 and 152-153**).

The 15 compounds with the highest activation of the GRHL reporter (i.e., GRHL^{WT-LUC}), as already stated explicitly in the original manuscript, are the ones listed in Fig. 2b and Fig. 3a. We have added an additional reference to Fig. 2b upon first mentioning of these 15 compounds (**line 159**). We would leave it to the discretion of the Editor whether it is helpful to also list these compounds in the text.

3) Please elaborate experimental conditions regarding generation of transgenic *C. elegans* (e.g. what concentration of pDEST-MB14 vector was used for microparticle bombardment?) or heat-shock conditions.

Authors’ Response:

We have now clarified these experimental conditions in the corresponding paragraphs of the Methods section (**lines 442-486, 569-570, 773-776**): “The pDEST-MB14 vectors for overexpression of *grh-1* under control of the endogenous *grh-1* promoter or the *hsp-16.2* promoter were transformed into the *unc-119*-deficient *C. elegans* strain HT1593 (*unc-119(ed3)* III) by microparticle bombardment using the biolistic particle delivery system PDS-1000/He (Bio-Rad) according to the manufacturer’s instructions and previously described protocols⁶¹, **with in total 7 µg of vector DNA per bombardment**. Homozygous transgenic lines with stable integration of the respective vector were selected based on GFP fluorescence (for *hsp-16.2* promoter constructs, selection by GFP was performed after transient, one hour heat shock at 33 °C) and backcrossed at least four times against wild-type N2 nematodes before being used in any experiments.” (...) “The following *C. elegans* strains used for this publication were provided by the *Caenorhabditis Genetics Center* (CGC, University of Minnesota, USA): Wild-type N2 (Bristol), RB754 *aak-2* (ok524), TJ1052 *age-1* (hx546), CB1370 *daf-2* (e1370), CF1038 *daf-16* (mu86), VC2072 *grh-1* (gk960), RB1813 *piki-1* (ok2346), EU31 *skn-1* (zu135). Strains overexpressing *grh-1* were generated as detailed below. For maintenance, nematodes were grown on Nematode Growth Medium (NGM) agar plates in 90 mm petri dishes at 20 °C using *E. coli* OP50 bacteria as a food source⁵⁹. **For experiments involving transient heat shock, nematodes on first day of adulthood were transferred to 33 °C for one hour and then returned to 20 °C, with the respective wild-type control nematodes subjected to the same procedure.** NGM agar

plates, after pouring, were dried at room temperature for 1-2 days and then stored at 4 °C until further use.”

4) Please explicitly mention the sample size bigger than three, although the authors present “sample sizes as indicated by depicted individual data points.”

Authors’ Response:

Sample sizes are now explicitly stated in the corresponding Figure Legends, where applicable. The Information for Authors specifically for Figures in *Nature Communications*, however, strongly endorse the use of individual data points, instead of summarizing bar graphs, which we preferred to adhere to in the manuscript.

5) In figure 1d, the authors need to show *C. elegans* fluorescence image into separate panels and magnify images to show nuclear localization of GRH-1 more clearly.

Authors’ Response:

We have now added the individual brightfield and 488 nm fluorescence microscopy images related to Fig. 1d as **newly added Supplementary Fig. 1a** and those related to Fig. 1g as **newly added Supplementary Fig. 1d**.

However, this again appears to be a very general misunderstanding. As already explicitly stated in the original manuscript, there is in fact no detectable nuclear GFP signal in *grh-1* OEx without heat shock, i.e., no signal over background and as compared to wild-type control nematodes without a recombinant GFP-tagged protein. Also see **lines 88-89** in the revised manuscript.

For the reviewer’s interest, we have included JPG of the respective panels to this Response to Reviewers file. Unlike for the PDF versions of the manuscript figures, these JPGs can be enlarged if required/desired.

6) Figure 3a appears to be more suitable in figure 2 than in figure 3. Please move figure 3a to figure 2.

Authors' Response:

Before submission of the initial version of the manuscript the Authors have discussed in length where to place Figure 3a, since it summarizes the individual lifespan data shown in Fig. 2c-q, but it also serves as a reference point for the subsequent panels in Fig. 3.

Since Fig. 2 contains the individual data/lifespans, however Fig. 3 requires the ranking of compounds' effects on lifespan for clarity, we opted back then to have Fig. 3a at the current location. Since this rationale has not changed significantly, we would prefer to maintain this panel at its previous location.

7) In Fig. 4b, please visualize Venn diagram to correctly describe the experimental scheme as GRHL1^{WT-TAG}-interacting proteins detected in both DMSO- and papaverine-treated conditions.

Authors' Response:

We thank Reviewer #1 for pointing out that the Venn diagram in Fig. 4b was not accurately reflecting the experimental scheme for identification of GRHL1^{WT-TAG}-interacting proteins and have changed it accordingly.

8) For Fig. 4c, it will be better to show whether the 7 PTM sites are conserved in *C. elegans* as well. In particular K116. If not, they need to discuss what that means in the discussion.

Authors' Response:

The functional role of the PTM sites is conserved in *C. elegans*, as indicated by the knock-down experiment depicted in Fig. 4h; however, the exact location of the sites appear to not be conserved in *C. elegans*, despite our significant efforts to identify the corresponding locations. As requested by the reviewer, we have expanded the Discussion section accordingly (lines 409-412).

9) In Fig. 4e, instead of darker edge colors, using different thickness of edges will be more intuitive.

Authors' Response:

As suggested by Reviewer #1 we have now, in addition to the intensity color-coding, increased edge thickness of stronger connections in Fig. 4e to make this panel visually more intuitive.

10) Please carefully distinguish *C. elegans* *daf-2* mutations, which confer many beneficial effects, from impaired glucose tolerance in humans.

Authors' Response:

We agree with the reviewer that *daf-2* impairment by RNAi, as well as incomplete loss-of-function mutations of *daf-2* extend lifespan in nematodes, whereas partial impairment of the insulin receptor in mice does not, as we have recently shown (Nature Communications, 2020, PubMedID 32350271). By contrast, and as also shown by us, the glucometabolic effect of *daf-2* impairment is very similar to impaired insulin signaling in mammalian cells (Cell Metabolism, 2012, PubMedID 22482728), i.e., both lead to impaired glucose transport/uptake. We have now further outlined this general consideration in the revised version of the manuscript (lines 454-466).

11) Please differently mark individual points in Supplementary Fig. 2a as in Supplementary Fig. 2d. In Supplementary Fig. 2c legend, use “gene set enrichment analysis”, not “gene sets enrichment analysis.”

Authors’ Response:

Assuming that Reviewer #1 is asking to mark individual points in Supplementary Fig. 2a similar to Supplementary Fig. 2c (and not similar to Supplementary Fig. 2d, which is not readily comparable), we have now changed Supplementary Fig. 2a to indicate GO terms corresponding to the individual points. However, please note:

- GO terms in Supplementary Fig. 2a and GO terms/pathways in Supplementary Fig. 2c do not match 100% and are therefore symbol-coded individually within each figure. Also see the next point-by-point response related to this.
- Several GO terms in Supplementary Fig. 2a overlap and hence not every single point is clearly visible. Since focus is on the MAPK pathway as the most significantly enriched, we believe this is negligible. Detailed information is now provided as part of the Source Data.

The legend of Supplementary Fig. 2c has been changed as suggested.

12) It will be better to perform GSEA of upregulated genes in *grh-1* OEx under conditions related to human *GRHL1* among different tissues to better compare the relationship between *C. elegans* *GRH-1* and human *GRHL1*.

Authors’ Response:

Supplementary Fig. 2a shows GO pathways enriched among genes found upregulated in *grh-1* OEx vs. wild-type control nematodes, using RNA extracted from whole animals (i.e., from a mixture of all cell and tissue types of *C. elegans*). With this approach, and generally using *C. elegans* as a model that does not readily allow dissection of individual tissues, it is not possible to assess gene expression in each individual nematodal tissue. Therefore, a direct tissue-by-tissue comparison of genes found upregulated together with *grh-1* in nematodes, to genes positively correlating with human *GRHL1* in individual human tissues is not feasible.

Instead, the analysis in Supplementary Fig. 2c used tissue-specific correlation coefficients between the expression of human genes and human *GRHL1* expression across all non-gender-specific tissues. These correlation coefficients were then used to rank the genes correlating with *GRHL1* in each human tissue and to determine enriched gene sets/pathways by GSEA. For clarity and as Reviewer #1 appears to also request in their comment, Supplementary Fig. 2c only depicts significant pathways correlated with human *GRHL1* that are identical or closely correspond to pathways also enriched among genes upregulated in *C. elegans* *grh-1* OEx. We find this to be the scientifically most unbiased approach to probe whether there likely is functional conservation in terms of global transcriptional regulation impacted by Grainyhead transcription factors across species. If we understand Reviewer #1 correctly, Supplementary Fig. 2c therefore should in fact already closely reflect what they are asking for in their comment. We have now tried to better explain in the Results what is depicted in Supplementary Fig. 2c (lines 124-128, 131).

14) In Supplementary Fig. 3 legend, use (a), not (A).

Authors’ Response:

This erroneous capitalization has been corrected.

15) In Supplementary Table 3, please mark which genes are related to MAP kinase activity and annotate gene name in the “FIMO - MEME Suite” table.

Authors’ Response:

This information has now been added to **Supplementary Data 3** (formerly Supplementary Table 3).

16) The authors mentioned that “The strongest activation of the GRHL reporter across the range of concentrations tested was observed with the compounds 2',4'-Dihydroxy-4-methoxychalcone (DHM-142 chalcone, 286.2 ± 36.6 at $20 \mu\text{M}$, $P = 0.008$) and alantolactone (131.5 ± 5.6 at $20 \mu\text{M}$, $P = 0.017$)” for describing Fig. 2b. However, it seems that vorinostat elicited the greatest effects across the range of concentrations. The authors should elaborate this sentence to better describe the data.

Authors’ Response:

Thank you for bringing this to our attention, this point was indeed worded unclear in the original manuscript. We thus have clarified this point and now also state the highest activation for vorinostat in the text and better explain the differences in efficacy and potency between the different compounds (**lines 164-170**). We also corrected a minor mistake in the SEM for alantolactone in the same section (should read “ 131.5 ± 6.8 ” not “ 131.5 ± 5.6 ”) that was noticed during assembling and checking the Source Data.

Reviewer #2 (Remarks to the Author):

Grigolon et al describe a study of the *C. elegans* transcription factor GRH1 (human GRHL1). They report the effects of a panel drugs upon GRH1 activation, identify interacting proteins and some PTMs, and provide evidence for links to protein regulation by methylation/demethylation and modulation by insulin/IGF-1 signaling network. Overall, this is a very well described study and results that provides novel information on aging and metabolism.

The differential GRH1 protein interaction results leads to the conclusion that protein methyl transferases (KMTs) and demethylases (KMDs) might be implicated in the regulation of GRH1. It is common practice to pursue reverse-interactome experiments to confirm the key interacting proteins. However, this was not attempted here. The authors should explain why this obvious experiment was omitted. (The authors do mention that KMT2C and KMT2D were found to interact with GRHL2, but this is not a proof in the context of the present study).

Authors' Response:

We thank Reviewer #2 for the appreciation of the manuscript as submitted, as well as this excellent suggestion and fully concur with the notion that this type of experiment is common practice to validate the physical interaction of individual proteins.

We did in fact repeatedly attempt this experimental approach but unfortunately were unsuccessful in obtaining stable cell lines overexpressing tagged KDM1A and KMT2C/KMT2D sufficiently suited for tandem-affinity purification and subsequent mass spectrometry, i.e., similar as performed for GRHL1, to further validate their interaction with GRHL1 in this way (i.e., by performing reverse-interactome experiments).

As a parallel option, we pursued an alternative (and in our hands experimentally more successful) route to probe in particular the functional interaction of methylases and demethylases with GRHL1 and papaverine in human cells (see Fig. 4f,g) and of demethylases with the papaverine-mediated lifespan-extension in *C. elegans* (see Fig. 4h), as already described in the original manuscript. We hope these experiments are considered sufficient in the context of the present study to support our claims that enzymes regulating lysine methylation impact Grainyhead transcription factor activity and nematodal lifespan-extension elicited by papaverine.

We would further like to point out that our differential proteomics approach (as outlined in Fig. 4b) utilized control samples derived from both DMSO and papaverine-treated HEK293 wild-type cells as stringent background controls. These control samples were subjected to the same tandem-affinity purification procedure and subsequent mass spectrometry analysis as the target samples, to thereby identify as potential GRHL1-interacting proteins only those that were fully absent from or enriched at least ≥ 3 -fold over the respective control samples (Methods section: "Proteins were considered as GRHL1 interactors if present in at least 2/3 samples GRHL1^{WT-TAG} and either fully absent or enriched at least ≥ 3 -fold over the respective control, based on sample average of unique peptide counts"). While this approach does not readily allow to discern direct from secondary/indirect interaction partners of a purified target protein, it greatly limits the false positive identification of particularly abundant and thus non-specifically co-purified proteins from a complex sample, and also of other proteins that non-specifically bind the affinity matrices utilized for the purification procedure.

In total 10 proteins known as involved in regulating lysine methylation were identified as potential GRHL1-interacting proteins over the background control samples, using our stringent experimental approach hopefully now further clarified and as already detailed in the Methods section of the

original manuscript. Notably, the lysine demethylase KDM1A in particular also represented one of the top co-purified proteins among all 130 potential GRHL1-interacting proteins. Given these results and together with the observation that GRHL1 is post-translationally modified by lysine methylation, we believe there can be little doubt of the functional relevance of this connection and subsequent focus on it.

We also would like to clearly make the point that the LC-MS based identification of GRHL1's potential interactors and post-translational modifications, in the context of the present study, served mainly as a first, albeit very important, step to select candidate interactors and post-translationally modified amino acids of GRHL1 for follow-up experiments. In these experiments, we used alternative and approaches to further probe the functional relevance of lysine methylation as a pathway affecting activity of GRHL1, which we believe is fully supported as a scientifically valid conclusion by the results of these investigations.

Nevertheless, to acknowledge this very valid point made by Reviewer #2, we have adapted wording in the corresponding Results paragraph (lines 216-225) and expanded the Discussion (lines 398-401) to more clearly and explicitly reflect that all GRHL1-interacting proteins here identified should only be considered as *potential* physical interaction partners until further experimental validation, to reflect a potential limitation of the current study.

One key aspect of the study is the putative regulation of the GRH1 by PTMs, particularly methylation. I acknowledge that the proteomics data is available via PRIDE, however, it is difficult to retrieve specific spectra of distinct peptides. I want to see the tandem mass spectra of the modified peptides as part of the supplementary materials. Please include figures of the annotated MS/MS spectra for all the modified peptides, including R9me2 and K116me2. Also, include the LC-MS traces (TIC) to demonstrate the quantitative differences between the unmodified and modified peptides, +/- papaverine. This data is relevant for establishing the role of methylation in regulation of GRH1.

Authors' Response:

As kindly requested by Reviewer #2, the annotated tandem mass spectra of GRHL1 peptides identified as post-translationally modified only after papaverine treatment, based on the identification results obtained using PEAKS X search engine, are now included as newly added **Supplementary Data 6**. We furthermore have clarified in the Results and Methods sections our approach to identify such peptides (lines 231-234 and 938-939).

Additionally, we have also included the extracted ion chromatogram (XIC) MS traces of the modified GRHL1 peptides from papaverine-treated and DMSO control samples (newly added **Supplementary Data 7**). The XICs of the corresponding unmodified peptides are provided only when these peptides were identified (3 out of the 7 modified peptide sequences). Note that, as now also explicitly stated in the manuscript (lines 234-238), upon more detailed inspection of the raw data we observed that GRHL1 generally was more abundant in papaverine-treated vs. DMSO control samples, and accordingly both modified and corresponding unmodified peptides increased after papaverine treatment. Thus, it is possible that certain modifications were identified exclusively in papaverine-treated samples due to the overall increase in GRHL1 abundance.

We fully acknowledge this point to be a relevant focus for future studies, i.e., to investigate in detail how papaverine might change GRHL1 protein levels or otherwise affect enrichment of this transcription factor from complex samples, e.g., through changes in GRHL1's protein interactome and/or GRHL1 protein folding and thereby altering accessibility of the purification tag.

Nevertheless, we do not think this detracts from the general conclusion that GRHL1 activity is impacted by lysine methylation. The various follow-up experiments already contained in the original manuscript (please see in particular Fig. 4d-h), for which the MS-based identification of potential GRHL1 interaction partners and PTMs served as the starting point, clearly indicate that post-translational lysine methylation is important to regulate the activity of Grainyhead transcription factors in humans, as well as in *C. elegans*, which is the main claim related to this in our current manuscript.

The affinity enrichment using biotin-papaverine is an elegant experiment (suppl. Fig 4). Is the binding mode of papaverine to proteins known? I wonder whether the position of biotin relative to papaverine will play a role, i.e. are there steric hindrances that might affect the way that proteins recognize/bind to papaverine? A short comment on this in the paper will suffice.

Authors' Response:

We thank Reviewer #2 for this -again- excellent and insightful comment. To our knowledge, some computational docking simulations aside (e.g., PMID 19336898), the exact mode by which papaverine binds to proteins is currently unknown. Hence, while steric hindrances following attachment of biotin to papaverine certainly might affect papaverine-protein interactions, we can only generally comment on this point. We have now added a comment regarding this in the Discussion (**lines 421-429**).

Please note that we have **now also added detailed Supplementary Methods describing the synthesis of biotinylated papaverine, including full NMR spectral data providing necessary proof of substance identity (especially the site of attachment of the biotinylated side chain).**

Reviewer #3 (Remarks to the Author):

In the provided manuscript, Grigolon et al. follow up on their previous finding that the activity of the transcription factor grainyhead 1 is a transcription factor controlling longevity. They show that overexpression of GRHL1 is sufficient to extend lifespan (assuming a mistake in Fig1h).

Authors' Response:

The Authors thank Reviewer #3 for this comment, which in part resembles a similar comment by Reviewer #1. We would like to clarify that Fig. 1e depicts an extended lifespan of the *grh-1* OEx strain **line 1** and now Supplementary Fig. 1c an extended lifespan of the independent *grh-1* OEx strain **line 2**, in both cases in the absence of any heat shock, i.e., at a continuous temperature of 20 °C. Even in the absence of a heat shock, and due to so-called leaky activity of the *hsp16.2* promoter, there is a limited (~2-3 fold) overexpression of *grh-1* (Fig. 1c and now Supplementary Fig. 1b).

As already described in the original manuscript and hopefully now clarified in the revised version (**lines 93-99**), Fig. 1h and now Supplementary Fig. 1f instead depict lifespan assays of the *grh-1* OEx strain lines 1 and 2 after transient heat shock, which increases *grh-1* overexpression massively (Fig. 1f and Supplementary Fig. 1e) and shortens lifespan, thus identifying the effect of *grh-1* overexpression on lifespan as dose-dependent.

We have now also added newly obtained data for a *grh-1* overexpressing strain in which expression is driven by its endogenous *grh-1* promoter (**newly added Supplementary Fig. 1h, i**). As now also stated in the corresponding Results paragraph, this strain further supports dose-dependency of the effect of *grh-1* overexpression, i.e., only limited overexpression being beneficial while intermediate or strong overexpression is not (**lines 102-105**).

Please also note the explanation in the point-by-point response to the minor comment #3 on the next page that Fig. 1h and now Supplementary Fig. 1f indeed display correct data from individual *grh-1* OEx strain lines.

They then conduct a cell culture-based chemical screen using a repurposing library. They identify inducers of GRHL1 expression and identify vorinostat, papaverine, and a series of other compounds. Consistent with their hypothesis that *grh1* is a regulator of longevity, some of the identified hits indeed extend lifespan and, most importantly, depend on GRHL1. They then show that these longevity compounds change post-translational modifications on GRHL1. Furthermore, the transcriptional activity of GRHL1 depends on methylation and methyltransferases. Finally, they correlate their findings to mouse and human gene expression datasets suggesting that GRHL1 may indeed be a valuable drug target to treat age-related diseases.

The manuscript is a good example of cross-species drug discovery and how chemicals can effectively work across multiple experimental models to identify evolutionarily conserved mechanisms modulating aging. The authors use a wide variety of different technologies to investigate the underlying mechanisms. The work is thoroughly done. I believe that this manuscript will be of general interest, both to researchers working in the field of aging and to researchers working in drug discovery. The manuscript is clearly written, well thought out, and easy to understand.

There are, however, a few minor issues that have to be addressed or answered. Once these are addressed, it will make an interesting and exciting publication.

Major:

- I don't think there are any major issues.

Authors' Response:

The authors highly appreciate the enthusiastic and positive comments of the reviewer, and particularly the absence of any major requests or criticisms.

Minor:

- Line 55: please indicate the *C.elegans* designation for KDM1A and KMT2C to make it easier for both mammalian and worm geneticists to read and connect to their field of study.

Authors' Response:

The Authors thank Reviewer #3 for this comment. We have now added the appropriate designations of the *C. elegans* KDM1A (LSD-1 and SPR-5) and KMT2D (SET-16; note that KMT2C at that point in the original manuscript was a typo and has been corrected to KMT2D) orthologues in the Introduction, and also for PIK3C2A (PIKI-1) mentioned in the same context (lines 61-63).

- Fig. 4h. did the authors try each demethylates individually? Are both at the same time necessary? Please clarify

Authors' Response:

We only performed preliminary experiments with RNAi against individual demethylases, specifically *lsd-1* alone. In these, papaverine appeared to still be able to extend lifespan, however to a reduced extent, consistent with additive activities of more than one demethylase.

Since both LSD-1 and SPR-5 represent similarly high confidence nematodal orthologues of human KDM1A, we opted instead to focus on the simultaneous knockdown of both corresponding genes at the same time. We have now clarified in the manuscript that both genes (i.e., *lsd-1* and *spr-5*) encode high confidence nematodal KDM1A orthologues (lines 273-274).

- The lifespan curves for the +HS in Figure 1e and + HS and supplementary Figure 1e look extremely similar. Please confirm that this is not an error. Since they are so similar it could be that the same lifespan curve was accidentally used for both figures but labeled to represent two different overexpression lines.

Authors' Response:

The Authors thank Reviewer #3 for this comment and assume they refer to Fig. 1h (not Fig. 1e) in the main manuscript and Supplementary Fig. 1e (in the original version, now Supplementary Fig. 1f).

We hereby confirm that the lifespan curves depicted in Fig. 1h (*grh-1* OEx strain line 1+ heat shock vs. wild-type control nematodes + heat shock) and now Supplementary Fig. 1f in the revised version (*grh-1* OEx strain line 2 + heat shock vs. wild-type control nematodes + heat shock), despite looking very similar, indeed do depict individual outcomes for the *grh-1* OEx strain lines 1 and 2. We further

note that these experiments used the same heat shocked wild-type control nematodes, as they were performed in parallel (see now Supplementary Data 1 for detailed lifespan assay statistics).

- Table S2 only reveals upregulated genes in the *grh-1* OE strain. The manuscript never mentions where the entire data set is available or how it will be made available?

Authors' Response:

The Authors thank Reviewer #3 for this comment. The original manuscript did state in the Methods section that the RNA-Seq data set was deposited in the NCBI's Gene Expression Omnibus, GEO Series accession number GSE159077. This is now again explicitly stated in the Data Availability statement of the revised version.

Reviewer access to these data is available upon request, however and independently, all data will be made publicly available upon publication of the study.

- Line 102 states that *pmk-1* is the highest expressed gene from the MAP kinase pathway. However, table S2 shows that *pmk-3* is expressed higher (1.078). So, is *pmk-3* not in the MAP kinase pathway?

Authors' Response:

The Authors thank Reviewer #3 for this comment. The original manuscript correctly states that *pmk-1* is "the most significantly upregulated gene [...]" of the MAP kinase activity pathway, i.e., most significantly overexpressed by *P*-value. Nevertheless, Reviewer #3 is correct in pointing out that *pmk-1* it is not the highest expressed gene from the MAP kinase activity pathway by log2 fold-change, since *pmk-3* is indeed also in the same pathway.

We have now clarified this in the revised manuscript (lines 119-122). Note that we have also corrected a related minor mistake in Supplementary Fig. 2b (placement of *pmk-3* in the Venn diagram).

- Table S3 contains binding sites for 102 genes for *grh-1*. The gene list S3 completely overlaps with the 242 upregulated genes in S2. That suggests that S3 was constructed by only searching for binding sites in upregulated genes. If that is correct, it should be stated to enable the reader to interpret the list.

Authors' Response:

The Authors thank Reviewer #3 for this comment. Reviewer #3 is absolutely correct in assuming how Supplementary Table 3 of the original manuscript (now Supplementary Data 3) was constructed. We indeed only scanned promoter regions of genes overexpressed in *grh-1* OEx for GRH-1 binding sites, to thereby gauge which of the upregulated genes are likely direct targets of GRH-1 transcriptional regulation. We apologize if this was unclear since it was only explicitly mentioned in the Methods section of the original manuscript.

This point is now clarified in the revised manuscript (lines 121-122).

- Table S5 (interactome) and Table S7 (papaverine pull-down) share a CPVL as a common component linking papaverine directly to GRHL1. It seems to me that this may be an exciting find. Any specific reason on why this was not followed up?

Authors' Response:

The Authors thank Reviewer #3 for this very astute comment. The specific reason why we did not follow up on this particular connection in the context of the current manuscript is its focus on post-translational modifications impacting GRHL1 activity, for which the methylases and demethylases identified as potential GRHL1-interacting proteins and their link to PIK3C2A as a potential papaverine interaction partner were most relevant. However, we fully recognize that the papaverine-CPVL-GRHL1 axis is very interesting to further explore and are in the process of setting up appropriate experiments. This point is now highlighted in the Discussion (**lines 433-437**).

Reviewers' Comments:

Reviewer #1:

Remarks to the Author:

The authors addressed my concerns adequately.

Reviewer #2:

Remarks to the Author:

The authors have revised the manuscript according to reviewers comments.

Reviewer #3:

Remarks to the Author:

The authors have answered all my concerns sufficiently and I support publication of the manuscript.